# Understanding AdamW through Proximal Methods and Scale-Freeness

**Zhenxun Zhuang**                                                        *zxzhuang@bu.edu*
*Boston University*

**Mingrui Liu**                                                          *mingruil@gmu.edu*
*George Mason University*

**Ashok Cutkosky**                                                    *ashok@cutkosky.com*
*Boston University*

**Francesco Orabona**                                              *francesco@orabona.com*
*Boston University*

**Reviewed on OpenReview:** *https://openreview.net/forum?id=IKhEPWGdwK*

## Abstract

Adam has been widely adopted for training deep neural networks due to less hyperparameter tuning and remarkable performance. To improve generalization, Adam is typically used in tandem with a squared $\ell_2$ regularizer (referred to as Adam-$\ell_2$). However, even better performance can be obtained with AdamW, which decouples the gradient of the regularizer from the update rule of Adam-$\ell_2$. Yet, we are still lacking a complete explanation of the advantages of AdamW. In this paper, we tackle this question from both an *optimization* and an *empirical* point of view. First, we show how to re-interpret AdamW as an approximation of a proximal gradient method, which takes advantage of the closed-form proximal mapping of the regularizer instead of only utilizing its gradient information as in Adam-$\ell_2$. Next, we consider the property of "scale-freeness" enjoyed by AdamW and by its proximal counterpart: their updates are invariant to component-wise rescaling of the gradients. We provide empirical evidence across a wide range of deep learning experiments showing a correlation between the problems in which AdamW exhibits an advantage over Adam-$\ell_2$ and the degree to which we expect the gradients of the network to exhibit multiple scales, thus motivating the hypothesis that the advantage of AdamW could be due to the scale-free updates.

## 1 Introduction

Recent years have seen a surge of interest in applying deep neural networks (LeCun et al., 2015) to a myriad of areas (Krizhevsky et al., 2012; Goodfellow et al., 2014; Vaswani et al., 2017; Wu et al., 2020). While Stochastic Gradient Descent (SGD) (Robbins & Monro, 1951) remains the dominant method for optimizing such models, its performance depends crucially on the step size hyperparameter. To alleviate this problem, there has been a significant amount of research on adaptive gradient methods (e.g. Duchi et al., 2010a; McMahan & Streeter, 2010; Tieleman & Hinton, 2012; Zeiler, 2012; Luo et al., 2018; Zhou et al., 2018; Zhang et al., 2018; Li & Orabona, 2019; 2020; Li et al., 2021). These methods provide mechanisms to automatically set stepsizes and have been shown to greatly reduce the tuning effort while maintaining good performance. Among these adaptive algorithms, one of the most widely used is Adam (Kingma & Ba, 2015), which achieves good results across a variety of problems even by simply adopting the default hyperparameter setting. Motivated by its huge successes, there has been much follow-up research addressing the theoretical convergence of Adam and its variants (Reddi et al., 2018; De et al., 2018; Zhou et al., 2018; Wang et al., 2020; Chen et al., 2019).

On the other hand, in practice, to improve the generalization ability, Adam is typically combined with a squared $\ell_2$ regularization, which we will call Adam-$\ell_2$ hereafter. Yet, as pointed out by Loshchilov & Hutter (2019), the gradient

of the regularizer does not interact properly with the Adam update rule. To address this, they provide a method called AdamW that decouples the gradient of the $\ell_2$ regularization from the update of Adam. The two algorithms are shown in Algorithm 1. Although AdamW is very popular (Kuen et al., 2019; Lifchitz et al., 2019; Carion et al., 2020) and it frequently outperforms Adam-$\ell_2$, it is currently unclear why it works so well. Recently, however, Bjorck et al. (2021) applied AdamW in Natural Language Processing and Reinforcement Learning problems and found no improvement of performance over sufficiently tuned Adam-$\ell_2$.

In this paper, we focus on understanding how the AdamW update differs from Adam-$\ell_2$ from an optimization point of view. First, we unveil the surprising connection between AdamW and *proximal updates* (Parikh & Boyd, 2014). In particular, we show that AdamW is an approximation of the latter and confirm such similarity with an empirical study. Moreover, noticing that AdamW and the proximal update are both *scale-free* while Adam-$\ell_2$ is not, we also derive a theorem showing that scale-free optimizers enjoy an automatic acceleration w.r.t. the condition number on certain cases. This gives AdamW a concrete theoretical advantage in training over Adam-$\ell_2$.

Next, we empirically identify the scenario of training very deep neural networks with Batch Normalization switched off as a case in which AdamW substantially outperforms Adam-$\ell_2$ in both testing and training. In such settings, we observe that the magnitudes of the coordinates of the updates during training are much more concentrated about a fixed value for AdamW than for Adam-$\ell_2$, which is an expected property of scale-free algorithms. Further, as depth increases, we expect a greater diversity of gradient scalings, a scenario that should favor scale-free updates. Our experiments support this hypothesis: deeper networks have more dramatic differences between the distributions of update scales between Adam-$\ell_2$ and AdamW and exhibit larger accuracy advantages for AdamW.

To summarize, the contributions of this paper are:

1. We show that AdamW can be seen as an approximation of a proximal update, which utilizes the entire regularizer rather than only its gradient.

2. We point out the scale-freeness property enjoyed by AdamW and show the advantage of such a property on a class of functions.

3. We find a scenario where AdamW is significantly better than Adam-$\ell_2$ in both training and testing performance and report an empirical observation of the correlation between such advantage and the scale-freeness property of AdamW.

The rest of this paper is organized as follows: In Section 2 we discuss the relevant literature. The connection between AdamW and the proximal updates as well as its scale-freeness are explained in Section 3. We then report the empirical observations in Section 4. Finally, we conclude with a discussion of the results, some limitations of this work, and future directions.

## 2 Related Work

**Weight decay** By biasing the optimization towards solutions with small norms, weight decay has long been a standard technique to improve the generalization ability in machine learning (Krogh & Hertz, 1992; Bos & Chug, 1996) and is still widely employed in training modern deep neural networks (Devlin et al., 2019; Tan & Le, 2019). Note that here we do not attempt to explain the generalization ability of weight decay or AdamW. Rather, we assume that the regularization and the topology of the network guarantee good generalization performance and study training algorithms from an optimization point of view. In this view, we are not aware of other work on the influence of regularization on the optimization process.

**Proximal updates** The use of proximal updates in the batch optimization literature dates back at least to 1965 (Moreau, 1965; Martinet, 1970; Rockafellar, 1976; Parikh & Boyd, 2014) and they were used in the online setting (Kivinen & Warmuth, 1997; Campolongo & Orabona, 2020), and also in the stochastic one (Toulis & Airoldi, 2017; Asi & Duchi, 2019). We are not aware of any previous paper pointing out the connection between AdamW and proximal updates.

**Scale-free algorithms** The scale-free property was first proposed in the online learning field (Cesa-Bianchi et al., 2007; Orabona & Pál, 2015; Orabona & Pál, 2018). There, they do not need to know a priori the Lipschitz constant of the functions, while obtaining optimal convergence rates. To the best of our knowledge, the connection between

---

**Algorithm 1** Adam with $\ell_2$ regularization (Adam-$\ell_2$) and AdamW Loshchilov & Hutter (2017).
*All operations on vectors are element-wise.*

---

1: **Given** $\alpha$, $\beta_1$, $\beta_2$, $\epsilon$, $\lambda \in \mathbb{R}$, lr schedule $\{\eta_t\}_{t\geq 0}$.
2: **Initialize:** $\boldsymbol{x}_0 \in \mathbb{R}^d$, $\boldsymbol{m}_0 \leftarrow 0$, $\boldsymbol{v}_0 \leftarrow 0$
3: **for** $t = 1, 2, \ldots, T$ **do**
4:     Compute a stochastic evaluation of the true gradient $\nabla f(\boldsymbol{x}_{t-1})$ denoted as $\nabla f_t(\boldsymbol{x}_{t-1})$
5:     $\boldsymbol{g}_t \leftarrow \nabla f_t(\boldsymbol{x}_{t-1})\ +\lambda \boldsymbol{x}_{t-1}$
6:     $\boldsymbol{m}_t \leftarrow \beta_1 \boldsymbol{m}_{t-1} + (1-\beta_1)\boldsymbol{g}_t$,    $\boldsymbol{v}_t \leftarrow \beta_2 \boldsymbol{v}_{t-1} + (1-\beta_2)\boldsymbol{g}_t^2$
7:     $\hat{\boldsymbol{m}}_t \leftarrow \boldsymbol{m}_t/(1-\beta_1^t)$,    $\hat{\boldsymbol{v}}_t \leftarrow \boldsymbol{v}_t/(1-\beta_2^t)$
8:     $\boldsymbol{x}_t \leftarrow \boldsymbol{x}_{t-1}\ -\eta_t \lambda \boldsymbol{x}_{t-1}\ -\eta_t \alpha \hat{\boldsymbol{m}}_t/(\sqrt{\hat{\boldsymbol{v}}_t} + \epsilon)$
9: **end for**

---

scale-freeness and the condition number we explain in Section 3 is novel, as is the empirical correlation between scale-freeness and good performance.

**Removing Batch Normalization (BN)** The setting of removing BN is not our invention: indeed, there is already active research in this (De & Smith, 2020; Zhang et al., 2019). The reason is that BN has many disadvantages (Brock et al., 2021) including added memory overhead (Bulò et al., 2018) and training time (Gitman & Ginsburg, 2017), and a discrepancy between training and inferencing (Singh & Shrivastava, 2019). BN has also been found to be unsuitable for many cases including sequential modeling tasks (Ba et al., 2016) and contrastive learning algorithms (Chen et al., 2020). Also, there are SOTA architectures that do not use BN including the Vision transformer (Dosovitskiy et al., 2021) and the BERT model (Devlin et al., 2019).

## 3 Theoretical Insights on Merits of AdamW

**AdamW and Proximal Updates** Here, we show that AdamW approximates a proximal algorithm (Moreau, 1965; Parikh & Boyd, 2014). A proximal algorithm is an algorithm for solving a convex optimization problem that uses the proximal operators of the objective function. The *proximal operator* $\text{prox}_h : \mathbb{R}^d \to \mathbb{R}^d$ of a convex function $h$ is defined for any $\boldsymbol{y} \in \mathbb{R}^d$ as $\text{prox}_h(\boldsymbol{y}) = \arg\min_{\boldsymbol{x}\in\mathbb{R}^d}(h(\boldsymbol{x}) + 1/2\|\boldsymbol{x} - \boldsymbol{y}\|_2^2)$.

Consider that we want to minimize the objective function

$$F(\boldsymbol{x}) = \tfrac{\lambda}{2}\|\boldsymbol{x}\|_2^2 + f(\boldsymbol{x}), \tag{1}$$

where $\lambda > 0$ and $f(\boldsymbol{x}) : \mathbb{R}^d \to \mathbb{R}$ is a function bounded from below. We could use a stochastic optimization algorithm that updates in the following fashion

$$\boldsymbol{x}_t = \boldsymbol{x}_{t-1} - \eta_t \boldsymbol{p}_t, \tag{2}$$

where $\eta_t$ is a learning rate schedule, e.g., the constant one or the cosine annealing (Loshchilov & Hutter, 2017) and $\boldsymbol{p}_t$ denotes any update direction. This update covers many cases, where $\alpha$ denotes the initial step size:

1. $\boldsymbol{p}_t = \alpha \boldsymbol{g}_t$ gives us the vanilla SGD;

2. $\boldsymbol{p}_t = \alpha \boldsymbol{g}_t/(\sqrt{\sum_{i=1}^t \boldsymbol{g}_i^2 + \epsilon})$ gives the AdaGrad algorithm (Duchi et al., 2010a);

3. $\boldsymbol{p}_t = \alpha \hat{\boldsymbol{m}}_t/(\sqrt{\hat{\boldsymbol{v}}_t} + \epsilon)$ recovers Adam (Kingma & Ba, 2015), where $\hat{\boldsymbol{m}}_t$ denotes the bias corrected first moment of past gradients and $\hat{\boldsymbol{v}}_t$ denotes the bias corrected second moment of past gradients as updated in Line 6-7 in Algorithm 1.

Note that in the above we use $\boldsymbol{g}_t$ to denote the stochastic gradient of the entire objective function: $\boldsymbol{g}_t = \nabla f_t(\boldsymbol{x}_{t-1}) + \lambda \boldsymbol{x}_{t-1}$ ($\lambda = 0$ if the regularizer is not present), where $\nabla f_t(\boldsymbol{x}_{t-1})$ is a stochastic evaluation of the true gradient $\nabla f(\boldsymbol{x}_{t-1})$.

This update rule (2) is given by the following online mirror descent update (Nemirovsky & Yudin, 1983; Warmuth & Jagota, 1997; Beck & Teboulle, 2003):

$$\boldsymbol{x}_t = \underset{\boldsymbol{x}\in\mathbb{R}^d}{\arg\min}\ \tfrac{\lambda}{2}\|\boldsymbol{x}_{t-1}\|_2^2 + f(\boldsymbol{x}_{t-1}) + \boldsymbol{p}_t^\top(\boldsymbol{x} - \boldsymbol{x}_{t-1}) + \tfrac{1}{2\eta_t}\|\boldsymbol{x} - \boldsymbol{x}_{t-1}\|_2^2. \tag{3}$$

| | Update (2) | Update (5) |
|---|---|---|
| $\boldsymbol{g}_t$ | $\nabla f_t(\boldsymbol{x}_{t-1}) + \lambda \boldsymbol{x}_{t-1}$ | $\nabla f_t(\boldsymbol{x}_{t-1})$ |
| $\boldsymbol{p}_t = \alpha \boldsymbol{g}_t$ | $\boldsymbol{x}_t = \boldsymbol{x}_{t-1} - \alpha \eta_t \boldsymbol{g}_t$ | $\boldsymbol{x}_t = \frac{1}{1+\lambda \eta_t}(\boldsymbol{x}_{t-1} - \alpha \eta_t \boldsymbol{g}_t)$ |
| $\boldsymbol{p}_t = \alpha \dfrac{\boldsymbol{g}_t}{\sqrt{\sum_{i=1}^t \boldsymbol{g}_i^2} + \epsilon}$ | $\boldsymbol{x}_t = \boldsymbol{x}_{t-1} - \alpha \eta_t \dfrac{\boldsymbol{g}_t}{\sqrt{\sum_{i=1}^t \boldsymbol{g}_i^2} + \epsilon}$ | $\boldsymbol{x}_t = \dfrac{1}{1+\lambda \eta_t}\left( \boldsymbol{x}_{t-1} - \alpha \eta_t \dfrac{\boldsymbol{g}_t}{\sqrt{\sum_{i=1}^t \boldsymbol{g}_i^2} + \epsilon} \right)$ |
| $\boldsymbol{p}_t = \alpha \dfrac{\hat{\boldsymbol{m}}_t}{\sqrt{\hat{\boldsymbol{v}}_t} + \epsilon}$ | $\boldsymbol{x}_t = \boldsymbol{x}_{t-1} - \alpha \eta_t \dfrac{\hat{\boldsymbol{m}}_t}{\sqrt{\hat{\boldsymbol{v}}_t} + \epsilon}$ | $\boldsymbol{x}_t = \dfrac{1}{1+\lambda \eta_t}\left( \boldsymbol{x}_{t-1} - \alpha \eta_t \dfrac{\hat{\boldsymbol{m}}_t}{\sqrt{\hat{\boldsymbol{v}}_t} + \epsilon} \right)$ |

Table 1: Comparison of Update (2) and (5) for different $\boldsymbol{p}_t$, where $\hat{\boldsymbol{m}}_t$ and $\hat{\boldsymbol{v}}_t$ are defined in Line 7 in Algorithm 1.

This approximates minimizing a first-order Taylor approximation of $F$ centered in $\boldsymbol{x}_{t-1}$ plus a term that measures the distance between the $\boldsymbol{x}_t$ and $\boldsymbol{x}_{t-1}$ according to the $\ell_2$ norm. The approximation becomes exact when $\boldsymbol{p}_t = \nabla f(\boldsymbol{x}_{t-1}) + \lambda \boldsymbol{x}_{t-1}$.

Yet, this is not the only way to construct first-order updates for the objective (1). An alternative route is to linearize only $f$ and to keep the squared $\ell_2$ norm in its functional form:

$$\boldsymbol{x}_t = \operatorname*{argmin}_{\boldsymbol{x} \in \mathbb{R}^d} \frac{\lambda}{2}\|\boldsymbol{x}\|_2^2 + f(\boldsymbol{x}_{t-1}) + \boldsymbol{p}_t^\top (\boldsymbol{x} - \boldsymbol{x}_{t-1}) + \frac{1}{2\eta_t}\|\boldsymbol{x} - \boldsymbol{x}_{t-1}\|_2^2 = \operatorname{prox}_{\frac{\lambda \eta_t}{2}\|\cdot\|_2^2}(\boldsymbol{x}_{t-1} - \eta_t \boldsymbol{p}_t), \qquad (4)$$

which uses the proximal operator of the convex function $\frac{\lambda \eta_t}{2}\|\cdot\|_2^2$. It is intuitive why this would be a better update: *We directly minimize the squared $\ell_2$ norm instead of approximating it.* We also would like to note that, similar to (3), the proximal updates of (4) can be shown to minimize the objective $F$ under appropriate conditions. However, we do not include the convergence analysis of (4) as this is already well-studied in the literature. For example, when $\boldsymbol{p}_t = \nabla f(\boldsymbol{x}_{t-1})$ in (4) and $f$ is convex and smooth, the update becomes a version of the (non-accelerated) iterative shrinkage-thresholding algorithm. This algorithm guarantees $F(\boldsymbol{x}_t) - F^* \leq O(1/t)$, which is in the same order as obtained by gradient descent on minimizing $f$ alone (Beck & Teboulle, 2009).

From the first-order optimality condition, the update is

$$\boldsymbol{x}_t = (1 + \lambda \eta_t)^{-1}(\boldsymbol{x}_{t-1} - \eta_t \boldsymbol{p}_t) . \qquad (5)$$

When $\lambda = 0$, the update in (2) and this one coincide. Yet, when $\lambda \neq 0$, they are no longer the same. For easier comparison between (2) and (5), we listed in Table 1 the detailed update formulas of them.

We now show how the update in (5) generalizes the one in AdamW. The update of AdamW is

$$\boldsymbol{x}_t = (1 - \lambda \eta_t)\boldsymbol{x}_{t-1} - \eta_t \alpha \hat{\boldsymbol{m}}_t / (\sqrt{\hat{\boldsymbol{v}}_t} + \epsilon) . \qquad (6)$$

On the other hand, using $\boldsymbol{p}_t = \alpha \hat{\boldsymbol{m}}_t / (\sqrt{\hat{\boldsymbol{v}}_t} + \epsilon)$ in (5) gives:

$$\boldsymbol{x}_t = (1 + \lambda \eta_t)^{-1}(\boldsymbol{x}_{t-1} - \eta_t \alpha \hat{\boldsymbol{m}}_t / (\sqrt{\hat{\boldsymbol{v}}_t} + \epsilon)), \qquad (7)$$

Its first-order Taylor approximation around $\eta_t = 0$ is

$$\boldsymbol{x}_t \approx (1 - \lambda \eta_t)\boldsymbol{x}_{t-1} - \eta_t \alpha \hat{\boldsymbol{m}}_t / (\sqrt{\hat{\boldsymbol{v}}_t} + \epsilon),$$

exactly the AdamW update (6). Hence, AdamW is a first-order approximation of a proximal update.

The careful reader might notice that the approximation from AdamW to the update in (7) becomes less accurate when $\eta_t$ becomes too large, and so be concerned whether this approximation is practical at all. Fortunately, in practice, $\eta_t$ is never large enough for this to be an issue. The remainder term of this approximation is $O(\lambda \eta_t^2)$ which we should always expect to be small as both $\lambda$ and $\eta_t$ are small. So, we can expect AdamW and the update in (7) to perform similarly for learning rate schedules $\eta_t$ commonly employed in practice, and we will indeed confirm this empirically in Section 4.3.

Let's now derive the consequences of this connection with proximal updates. First of all, at least in the convex case, the convergence rate of the proximal updates will depend on $\|\nabla f(\boldsymbol{x}_t)\|_2^2$ rather than on $\|\nabla f(\boldsymbol{x}_t) + \lambda \boldsymbol{x}_t\|_2^2$ (Duchi et al.,

2010b). This could be a significant improvement: the regularized loss function is never Lipschitz, so the regularized gradients $\nabla f(\boldsymbol{x}_t) + \lambda \boldsymbol{x}_t$ could be much larger than $\nabla f(\boldsymbol{x}_t)$ when $f$ itself is Lipschitz.

More importantly, proximal updates are fundamentally better at keeping the weights small. Let us consider a couple of simple examples to see how this could be. First, suppose the weights are *already zero*. Then, when taking an update according to (2), we increase the weights to $-\eta_t \boldsymbol{p}_t$. In contrast, update (5) clearly leads to a smaller value. This is because it computes an update using the regularizer rather than its gradient. As an even more disturbing, yet actually more realistic example, consider the case that $\boldsymbol{x}_{t-1}$ is non-zero, but $\boldsymbol{g}_t = \boldsymbol{0}$. In this case, taking an update using (2) may actually *increase* the weights by causing $\boldsymbol{x}_t$ to *overshoot* the origin. In contrast, the proximal update will never demonstrate such pathological behavior. Notice that this pathological behavior of (2) can be mitigated by properly tuning the learning rate. However, one of the main attractions of adaptive optimizers is that we should not need to tune the learning rate as much. Thus, *the proximal update can be viewed as augmenting the adaptive methods with an even greater degree of learning-rate robustness.*

**AdamW is Scale-Free** We have discussed what advantages the proximal step hidden in AdamW can give but have not yet taken into consideration the specific shape of the update. Here instead we will look closely at the $\boldsymbol{p}_t$ used in AdamW to show its *scale-freeness*. Our main claim is: *the lack of scale-freeness seems to harm Adam-$\ell_2$'s performance in certain scenarios in deep learning, while AdamW preserves the scale-freeness even with an $\ell_2$ regularizer*. We will motivate this claim theoretically in this section and empirically in Section 4.

An optimization algorithm is said to be *scale-free* if its iterates do not change when one multiplies any coordinate of all the gradients of the losses $f_t$ by a positive constant (Orabona & Pál, 2018). It turns out that the update (6) of AdamW and the update (7) are both scale-free when $\epsilon = 0$. This is evident for AdamW as the scaling factor for any coordinate of the gradient is kept in both $\hat{\boldsymbol{m}}_t$ and $\sqrt{\hat{\boldsymbol{v}}_t}$ and will be canceled out when dividing them. *(In practical applications, though, $\epsilon$ is very small but not zero, so we empirically verify in Section 4.2 that it is small enough to still approximately ensure the scale-free property.)* In contrast, for Adam-$\ell_2$, the addition of the weight decay vector to the gradient (Line 5 of Algorithm 1) destroys this property.

We want to emphasize the comparison between Adam-$\ell_2$ and AdamW: once Adam-$\ell_2$ adopts a non-zero $\lambda$, it loses the scale-freeness property; in contrast, AdamW enjoys this property for arbitrary $\lambda$. The same applies to any AdaGrad-type and Adam-type algorithm that incorporates the squared $\ell_2$ regularizer by simply adding the gradient of the $\ell_2$ regularizer directly to the gradient of $f$, as in Adam-$\ell_2$ (as implemented in Tensorflow and Pytorch). Such algorithms are scale-free only when they do not use weight decay.

Also, as we wrote above, AdamW can be seen as the first-order Taylor approximation on $\eta_t = 0$ of (7); in turn, the scale-freeness of (7) directly comes from the proximal updates. Of course, there may be other ways to design scale-free updates solving (1); yet, for AdamW, its scale-free property derives directly from the proximal update.

We stress that the scale-freeness is an important but largely overlooked property of an optimization algorithm. It has already been utilized to explain the success of AdaGrad (Orabona & Pál, 2018). Recently, Agarwal et al. (2020) also provides theoretical and empirical support for setting the $\epsilon$ in the denominator of AdaGrad to be 0, thus making the update scale-free.

Below, we show how scale-freeness can reduce the condition number of a certain class of functions.

**Scale-Freeness Provides Preconditioning** For a twice continuously differentiable function $f$, its Hessian matrix is symmetric and its *condition number* $\kappa$ is defined as the ratio of its largest absolute value eigenvalue to its smallest one. It is well-known that the best convergence rate when minimizing such $f$ using a first-order optimization algorithm (e.g., gradient descent) must depend on the condition number (Theorem 2.1.13, Nesterov, 2004). In particular, a problem with a small $\kappa$ can be solved more efficiently than one with a big $\kappa$. One way to reduce the effect of the condition number is to use a *preconditioner* (Nocedal & Wright, 2006). While originally designed for solving systems of linear equations, preconditioning can be extended to the optimization of non-linear functions and it should depend on the Hessian of the function (Boyd & Vandenberghe, 2004; Li, 2018). However, it is unclear how to set the preconditioner given that the Hessian might not be constant (Section 9.4.4 Boyd & Vandenberghe, 2004) and in stochastic optimization the Hessian cannot be easily estimated (Li, 2018).

In the following theorem, we show that scale-freeness gives similar advantages to the use of an optimal diagonal preconditioner, *for free* (proof in the Appendix). Specifically, a scale-free algorithm can automatically transform solving

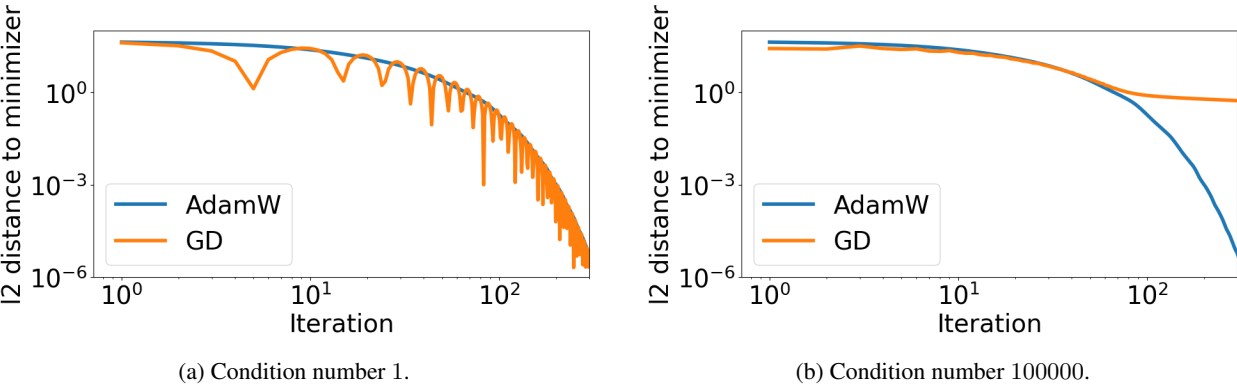

(a) Condition number 1.

(b) Condition number 100000.

Figure 1: Non-scale-free GD v.s. scale-free AdamW on quadratic functions with different condition numbers.

the original problem into solving a problem with a potentially much smaller $\kappa$ and thus could provide substantial improvements over non-scale-free ones, as shown in Figure 1 in the paper.

**Theorem 3.1.** *Let $f$ be a twice continuously differentiable function and $\boldsymbol{x}^*$ such that $\nabla f(\boldsymbol{x}^*) = \mathbf{0}$. Then, let $\tilde{f}_\Lambda$ be the family of functions such that $\nabla \tilde{f}_\Lambda(\boldsymbol{x}^*) = \mathbf{0}$, and $\nabla^2 \tilde{f}_\Lambda(\boldsymbol{x}) = \Lambda \nabla^2 f(\boldsymbol{x})$, where $\Lambda = diag(\lambda_1, \ldots, \lambda_d) \succeq 0$. Then, running any scale-free optimization algorithm on $f$ and $\tilde{f}_\Lambda$ will result exactly in the same iterates, assuming the same noise on the gradients. Moreover, any dependency on the condition number of the scale-free algorithm will be reduced to the smallest condition number among all the functions $\tilde{f}_\Lambda$.*

To give an example of when this is advantageous, consider when $\nabla^2 f(\boldsymbol{x})$ is a diagonal matrix:

$$\nabla^2 f(\boldsymbol{x}) = diag(g_1(\boldsymbol{x}), g_2(\boldsymbol{x}), \ldots, g_d(\boldsymbol{x})) .$$

Assume $0 < \mu \leq \mu_i \leq g_i(\boldsymbol{x}) \leq M_i \leq M$ for $i \in \{1, \ldots, d\}$. Denote $j = \arg\max_i M_i/\mu_i$. Choose $\lambda_i$ s.t. $\mu_j \leq \lambda_i\mu_i \leq \lambda_i g_i(\boldsymbol{x}) \leq \lambda_i M_i \leq M_j$ then $\Lambda\nabla^2 f(\boldsymbol{x})$ has a condition number $\kappa' = M_j/\mu_j$. This gives scale-free algorithms a big advantage when $\max_i M_i/\mu_i \ll M/\mu$.

Another example is one of the quadratic functions.

**Corollary 3.2.** *For quadratic problems $f(\boldsymbol{x}) = \frac{1}{2}\boldsymbol{x}^\top H\boldsymbol{x} + \boldsymbol{b}^\top \boldsymbol{x} + c$, with $H$ diagonal and positive definite, any scale-free algorithm will not differentiate between minimizing $f$ and $\tilde{f}(\boldsymbol{x}) = \frac{1}{2}\boldsymbol{x}^\top \boldsymbol{x} + (H^{-1}\boldsymbol{b})^\top \boldsymbol{x} + c$. As the condition number of $\tilde{f}$ is 1, the operation, and most importantly, the convergence, of a scale-free algorithm will not be affected by the condition number of $f$ at all.*

Figure 1 illustrates Corollary 3.2: we compare GD (non-scale-free) with AdamW (scale-free) on optimizing two quadratic functions with the same minimizer, but one's Hessian matrix being a rescaled version of the other's, resulting in different condition numbers. The figure clearly shows that, even after tuning the learning rates, the updates of AdamW (starting from the same point) and thus its convergence to the minimizer, is completely unaffected by the condition number, while GD's updates change drastically and its performance deteriorates significantly when the condition number is large. It is not hard to imagine that such poor training performance would likely also lead to poor testing performance.

This can also explain AdaGrad's improvements over SGD in certain scenarios. As an additional example, in Appendix B we analyze a variant of AdaGrad with restarts and show an improved convergence on strongly convex functions due to scale-freeness. Note that the folklore justification for such improvements is that the learning rate of AdaGrad approximates the inverse of the Hessian matrix, but this is incorrect: AdaGrad does not compute Hessians and there is no reason to believe it approximates them in general.

More importantly, another scenario demonstrating the advantage of scale-freeness is training deep neural networks. Neural networks are known to suffer from the notorious problem of vanishing/exploding gradients (Bengio et al., 1994; Glorot & Bengio, 2010; Pascanu et al., 2013). This problem leads to the gradient scales being very different across layers, especially between the first and the last layers. The problem is particularly severe when the model is not

equipped with normalization mechanisms like Batch Normalization (Ioffe & Szegedy, 2015). In such cases, when using a non-scale-free optimization algorithm (e.g., SGD), the first layers and the last layers will proceed at very different speeds, whereas a scale-free algorithm ensures that each layer is updated at a similar pace. We will investigate these effects empirically in the next section.

## 4 Deep Learning Empirical Evaluation

In this section, we empirically compare Adam-$\ell_2$ with AdamW. First (Section 4.1), we report experiments for deep neural networks on image classification tasks (CIFAR10/100). Here, AdamW enjoys a significant advantage over Adam-$\ell_2$ when BN is switched off on deeper neural networks. We also report the correlation between this advantage and the scale-freeness property of AdamW. Next (Section 4.2), we show that AdamW is still almost scale-free even when the $\epsilon$ used in practice is not 0, and how, contrary to AdamW, Adam-$\ell_2$ is not scale-free. Finally (Section 4.3), we show that AdamW performs similarly to the update in (7), which we will denote by AdamProx below, thus supporting the observations in Section 3.

**Data Normalization and Augmentation:** We consider the image classification task on CIFAR-10/100 datasets. Images are normalized per channel using the means and standard deviations computed from all training images. We adopt the data augmentation technique following Lee et al. (2015) (for training only): 4 pixels are padded on each side of an image and a $32 \times 32$ crop is randomly sampled from the padded image or its horizontal flip.

**Models:** For the CIFAR-10 dataset, we employ the Residual Network[1] model (He et al., 2016) of 20/44/56/110/218 layers; and for CIFAR-100, we additionally utilize the DenseNet-BC[2] model (Huang et al., 2017) with 100 layers and a growth-rate of 12. The loss is the cross-entropy loss.

**Hyperparameter tuning:** For both Adam-$\ell_2$ and AdamW, we set $\beta_1 = 0.9$, $\beta_2 = 0.999$, $\epsilon = 10^{-8}$ as suggested in the original Adam paper Kingma & Ba (2015). To set the initial step size $\alpha$ and weight decay parameter $\lambda$, we grid search over $\{0.00005, 0.0001, 0.0005, 0.001, 0.005\}$ for $\alpha$ and $\{0, 0.00001, 0.00005, 0.0001, 0.0005, 0.001\}$ for $\lambda$. Whenever the best performing hyperparameters lie in the boundary of the searching grid, we always extend the grid to ensure that the final best-performing hyperparameters fall into the interior of the grid.

**Training:** For each experiment configuration (e.g., 110-layer Resnet without BN), we randomly select an initialization of the model to use as a fixed starting point for all optimizers and hyperparameter settings. We use a mini-batch of 128, and train 300 epochs unless otherwise specified.

### 4.1 AdamW vs. Adam-$\ell_2$: Influence of Batch Normalization and Correlation with Scale-freeness

**With BN, Adam-$\ell_2$ is on par with AdamW** Recently, Bjorck et al. (2021) found that AdamW has no improvement in absolute performance over sufficiently tuned Adam-$\ell_2$ in some reinforcement learning experiments. We also discover the same phenomenon in several image classification tasks, see Figure 2. Indeed, the best weight decay parameter is 0 for all cases and AdamW coincides with Adam-$\ell_2$ in these cases. Nevertheless, AdamW does decouple the optimal choice of the weight decay parameter from the initial step size much better than Adam-$\ell_2$ in all cases.

**Removing BN** Notice that the models used in Figure 2 all employ BN. BN works by normalizing the input to each layer across the mini-batch to make each coordinate have zero-mean and unit-variance. Without BN, deep neural networks are known to suffer from gradient explosion and vanishing (Schoenholz et al., 2017). This means each coordinate of the gradient will have very different scales, especially between the first and last layers. For non-scale-free algorithms, the update to the network weights will also be affected and each coordinate will proceed at a different pace. In contrast, scale-free optimizers are robust to such issues as the scaling of any single coordinate will not affect the update. Thus, we consider the case where BN is removed as that is where AdamW and Adam-$\ell_2$ will show very different patterns due to scale-freeness.

**Without BN, AdamW Outperforms Adam-$\ell_2$** In fact, without BN, AdamW outperforms Adam-$\ell_2$ even when both are finely tuned, especially on relatively deep neural networks (see Figure 3 and 4). AdamW not only obtains a much better test accuracy but also trains much faster.

---

[1] https://github.com/akamaster/pytorch_resnet_cifar10
[2] https://github.com/bearpaw/pytorch-classification

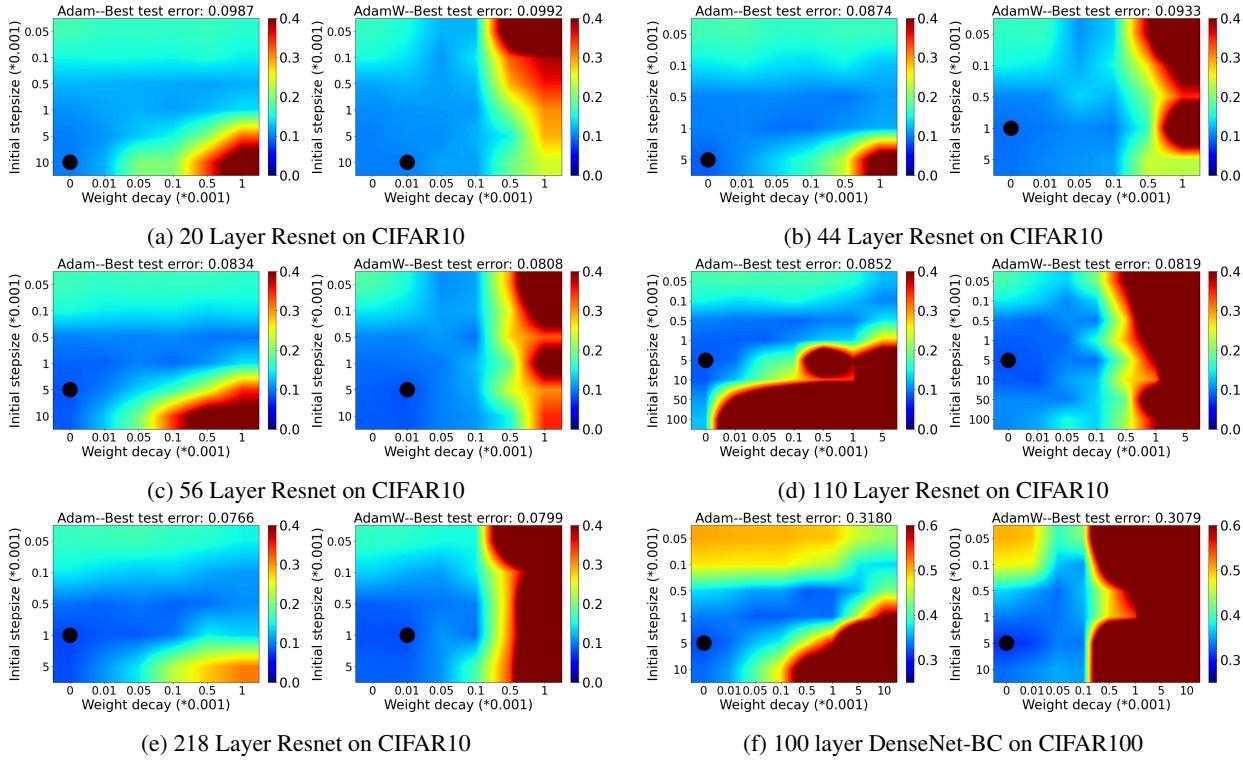

Figure 2: The final Top-1 test error on using AdamW vs. Adam-$\ell_2$ on training a Resnet/DenseNet with Batch Normalization on CIFAR10/100 (*the black circle denotes the best setting*). Note how close are the best performing hyperparameter combinations and the lowest testing error each optimizer obtains between Adam-$\ell_2$ and AdamW for each setting suggesting they perform similarly when BN is turned on.

**AdamW's Advantage and Scale-freeness** We also observe that the advantage of AdamW becomes more evident as the network becomes deeper. Recall that as the depth grows, without BN, the gradient explosion and vanishing problem becomes more severe. This means that for the non-scale-free Adam-$\ell_2$, the updates of each coordinate will be dispersed on a wider range of scales even when the same weight decay parameter is employed. In contrast, the scales of the updates of AdamW will be much more concentrated in a smaller range. This is exactly verified empirically as illustrated in the 5th & 6th columns of figures in Figure 3 and 4. There, we report the histograms of the absolute value of updates of Adam-$\ell_2$ vs. AdamW of all coordinates near the end of training (for their comparison over the whole training process please refer to the Appendix C).

This correlation between the advantage of AdamW over Adam-$\ell_2$ and the different spread of update scales which is induced by the scale-freeness property of AdamW provides empirical evidence on when AdamW excels over Adam-$\ell_2$.

**SGD and Scale-freeness** The reader might wonder why SGD is known to provide state-of-the-art performance on many deep learning architectures (e.g., He et al., 2016; Huang et al., 2017) *without* being scale-free. At first blush, this seems to contradict our claims that scale-freeness correlates with good performance. In reality, the good performance of SGD in very deep models is linked to the use of BN that normalizes the gradients. Indeed, we verified empirically that SGD fails spectacularly when BN is not used. For example, on training the 110 layer Resnet without BN using SGD with momentum and weight decay of $0.0001$, even a learning rate of $1e-10$ will lead to divergence.

## 4.2 Verifying Scale-Freeness

In the previous section, we elaborated on the scale-freeness property of AdamW and its correlation with AdamW's advantage over Adam-$\ell_2$. However, one may notice that in practice, the $\epsilon$ factor in the AdamW update is typically small but not 0, in our case $1e$-8, thus preventing it from completely scale-free. In this section, we verify that the effect of such an $\epsilon$ on the scale-freeness is negligible.

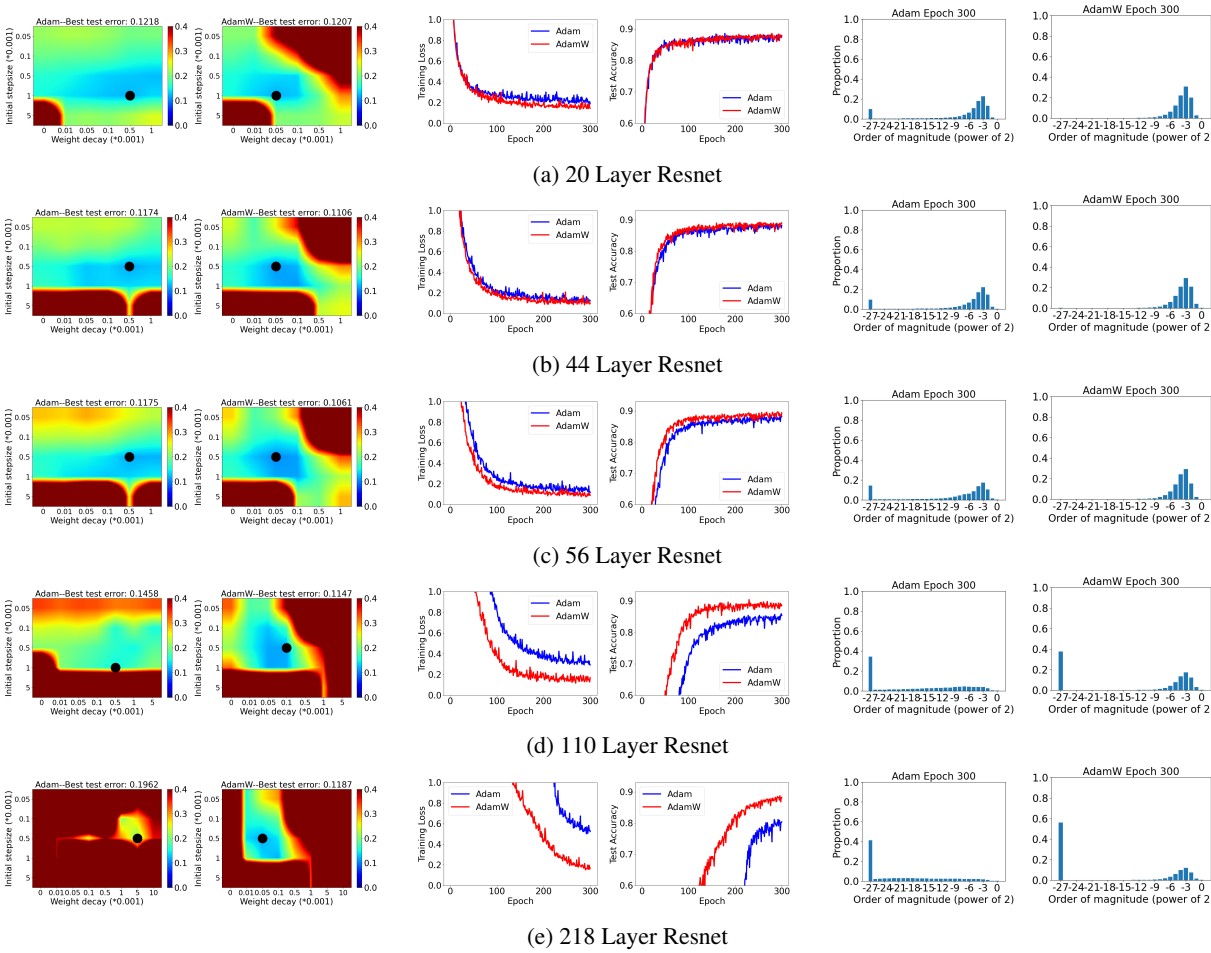

Figure 3: On using AdamW vs. Adam-$\ell_2$ on training a Resnet without Batch Normalization on CIFAR10. (Left two) The final Top-1 test error (*the black circle denotes the best setting*). (Middle two) The training loss and test accuracy curve when employing the initial step size and the weight decay parameter that gives the smallest test error. (Right two) The histogram of the magnitude of corresponding updates of all coordinates of the network near the end of the training when employing the initial step size and the weight decay parameter that gives the smallest test error. Note that as the depth of the neural network increases, Adam-$\ell_2$'s updates scatter more evenly over the entire spectrum while AdamW's updates are still concentrated in a small range, and AdamW's advantage in both training and testing over Adam-$\ell_2$ becomes more significant.

As a simple empirical verification of the scale-freeness, we consider the scenario where we multiply the loss function by a positive number. Note that any other method to test scale-freeness would be equally good. For a feed-forward neural network without BN, this means the gradient would also be scaled up by that factor. In this case, the updates of a scale-free optimization algorithm would remain exactly the same, whereas they would change for an optimization algorithm that is not scale-free.

Figure 5 shows results of the loss function being multiplied by 10 and 100 respectively on optimizing a 110-layer Resnet with BN *removed*. For results of the original loss see Figure 3d. We can see that AdamW has almost the same performance across the range of initial learning rates and weight decay parameters, and most importantly, the best values of these two hyperparameters remain the same. This verifies that, even when employing a (small) non-zero $\epsilon$, AdamW is still approximately scale-free. In contrast, Adam-$\ell_2$ is not scale-free and we can see that its behavior varies drastically with the best initial learning rates and weight decay parameters in each setting totally different.

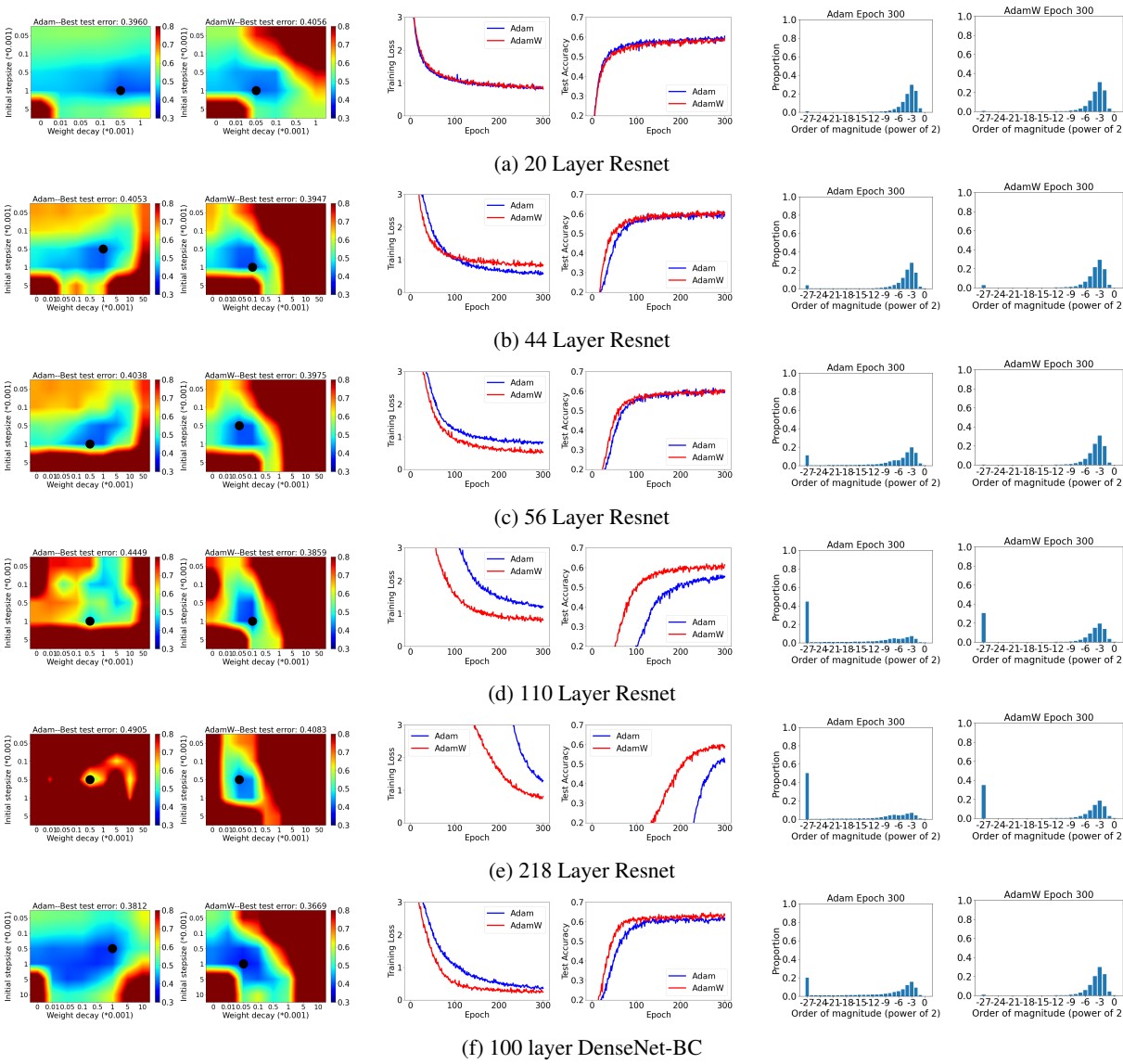

Figure 4: On using AdamW vs. Adam-$\ell_2$ on training a Resnet/DenseNet without Batch Normalization on CIFAR100. (Left two) The final Top-1 test error (*the black circle denotes the best setting*). (Middle two) The training loss and test accuracy curve when employing the initial step size and the weight decay parameter that gives the smallest test error. (Right two) The histogram of the magnitude of corresponding updates of all coordinates of the network near the end of the training when employing the initial step size and the weight decay parameter that gives the smallest test error. Note that as the depth of the neural network increases, Adam-$\ell_2$'s updates scatter more evenly over the entire spectrum while AdamW's updates are still concentrated in a small range, and AdamW's advantage in both training and testing over Adam-$\ell_2$ becomes more significant.

### 4.3 AdamW and AdamProx Behave very Similarly

In Section 3, we showed theoretically that AdamW is the first order Taylor approximation of AdamProx (update rule (7)). Beyond this theoretical argument, here we verify empirically that the approximation is good. In Figure 6, we consider the case when $\eta_t = 1$ for all $t$ - a relatively large constant learning rate schedule. In such cases, AdamW and AdamProx still behave very similarly. This suggests that for most learning rate schedules, e.g., cosine, exponential, polynomial, and step decay, which all monotonously decrease from $\eta_0 = 1$, AdamProx will remain a very good approximation to AdamW. Thus, it is reasonable to use the more classically-linked AdamProx to try to understand AdamW.

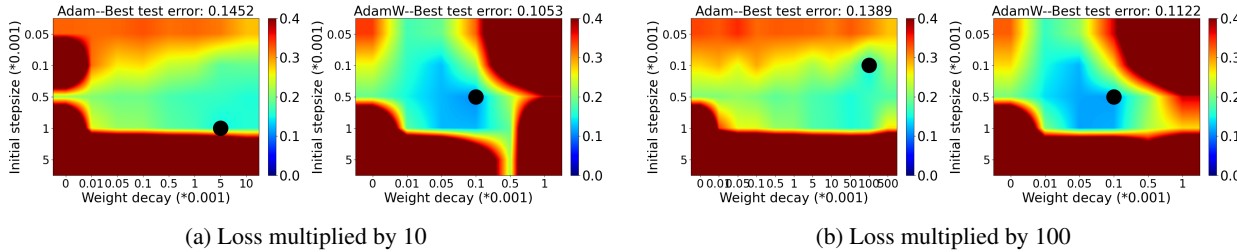

(a) Loss multiplied by 10          (b) Loss multiplied by 100

Figure 5: The final top-1 test error of AdamW vs. Adam-$\ell_2$ on optimizing a 110-layer Resnet with BN *removed* on CIFAR-10 with the loss function multiplied by 10 (left two figures) and 100 (right two figures). Note how the best performing hyperparameter combinations of AdamW remain the same for different loss multiplication factors as well as the shape of the heatmap being very similar. In contrast, Adam-$\ell_2$'s performance as well as the best performing hyperparameter combinations vary dramatically for different loss multiplication factors.

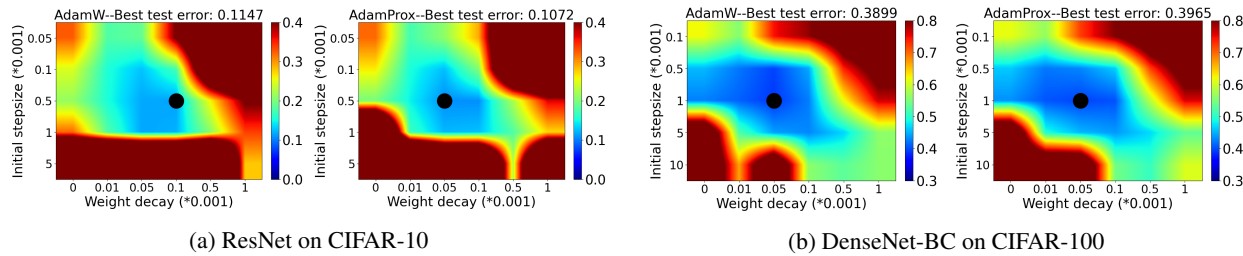

(a) ResNet on CIFAR-10          (b) DenseNet-BC on CIFAR-100

Figure 6: The final Top-1 test error of using AdamW vs. AdamProx on training (*the black circle denotes the best setting*). (Top row) a 110-layer ResNet with BN *removed* on CIFAR-10 (trained for 300 epochs). (Bottom row) a 100-layer DenseNet-BC with BN *removed* on CIFAR-100 (trained for 100 epochs). Note how similar are the shapes of the heatmaps, the best performing hyperparameter combinations, and the test errors between AdamW and AdamProx.

## 5 Conclusion and Future Work

In this paper, we provide insights for understanding the merits of AdamW from two points of view. We first show that AdamW is an approximation of the proximal updates both theoretically and empirically. We then identify the setting of training very deep neural networks without batch normalization in which AdamW substantially outperforms Adam-$\ell_2$ in both training and testing and show its correlation with the scale-freeness property of AdamW. Nevertheless, we are aware of some limitations of this work as well as many directions worth exploring.

**Limitations** First, we only focus on investigating the effects of scale-freeness on Adam-$\ell_2$ and AdamW, but it would be interesting to study scale-freeness more generally. Also, what we showed is just a correlation instead of causality thus we did not rule out other possible causes beyond scale-freeness for the success of AdamW. Indeed, rigorously proving causality for any such claim is extremely difficult - even in the hard sciences. Note that there are papers claiming that adaptive updates have worse generalization (Wilson et al., 2017); however, such claims have been recently partly con-

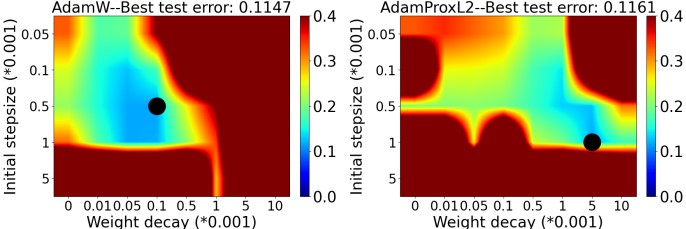

Figure 7: Final Top-1 test error of using AdamW vs. AdamProxL2 to train a 110-layer ResNet *without* BN on CIFAR10 (*the black circle denotes the best setting*).

futed (see, e.g., Agarwal et al., 2020). On this note, despite its empirical success, we stress that Adam will not even converge on some convex functions (Reddi et al., 2018), thus making it hard to prove formal theoretical convergence and/or generalization guarantees.

**Update for no-square $\ell_2$ regularization.** Instead of using the squared $\ell_2$ regularization, we might think to use the $\ell_2$ regularization, that is without the square. This is known to have better statistical properties than the squared $\ell_2$ (see,

e.g., Orabona, 2014), but it is not smooth so it is harder to be optimized. However, with proximal updates, we don't have to worry about its non-smoothness. Hence, we can consider the objective function $F(\boldsymbol{x}) = \lambda\|\boldsymbol{x}\|_2 + f(\boldsymbol{x})$.

The corresponding prox-SGD update was derived in Duchi & Singer (2009) for scalar learning rates and it is easy to generalize to our setting as

$$\boldsymbol{x}_{t+1} = \max\left(1 - \frac{\lambda\eta_t}{\|\boldsymbol{x}_t - \eta_t\boldsymbol{p}_t\|}, 0\right)(\boldsymbol{x}_t - \eta_t\boldsymbol{p}_t) \ .$$

Its performance, named AdamProxL2, as shown in Figure 7, can be on a par with AdamW.

**Distributed Training** Batch normalization is not user-friendly in distributed training as it requires each machine to collect a batch of statistics to update the model which may be inaccurate when machines do not communicate frequently with each other Goyal et al. (2017). Since AdamW outperforms Adam-$\ell_2$ significantly in settings without BN, at least in feed-forward neural networks, we can apply AdamW in distributed training to see if it still enjoys the same merits.

### Broader Impact Statement

The main contribution of this paper is the study of a known optimization algorithm AdamW from the theoretical angles of proximal updates and scale-freeness, while the experiments are done to empirically validate and support the theoretical findings. It is a general algorithm and we do not specify in which applications should it be employed, thus we do not foresee any direct negative societal impact our work might cause.

## Acknowledgments

Francesco Orabona is supported by the National Science Foundation under the grants no. 1908111 "AF: Small: Collaborative Research: New Representations for Learning Algorithms and Secure Computation", no. 2022446 "Foundations of Data Science Institute", and no. 2046096 "CAREER: Parameter-free Optimization Algorithms for Machine Learning". Mingrui Liu is supported in part by a grant from George Mason University.

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

# Appendices

## A  Proof of Theorem 3.1

*Proof.* From the Fundamental Theorem of Calculus we have:

$$\nabla f(\boldsymbol{x}) = \nabla f(\boldsymbol{x}^*) + \int_0^1 \nabla^2 f(\boldsymbol{x}^* + t(\boldsymbol{x} - \boldsymbol{x}^*))(\boldsymbol{x} - \boldsymbol{x}^*)dt = \int_0^1 \nabla^2 f(\boldsymbol{x}^* + t(\boldsymbol{x} - \boldsymbol{x}^*))(\boldsymbol{x} - \boldsymbol{x}^*)dt \ .$$

Thus, for any function $\tilde{f}_\Lambda(\boldsymbol{x})$ whose Hessian is $\Lambda \nabla^2 f(\boldsymbol{x})$ and $\nabla \tilde{f}_\Lambda(\boldsymbol{x}^*) = 0$, we have $\nabla \tilde{f}_\Lambda(\boldsymbol{x}) = \Lambda \nabla f(\boldsymbol{x})$.

Now, from the definition of a scale-free algorithm, the iterates of such an algorithm do not change when one multiplies each coordinate of all the gradients by a positive constant. Thus, a scale-free algorithm optimizing $f$ behaves the same as if it is optimizing $\tilde{f}_\Lambda$. □

## B  A Scale-free Algorithm with Dependency on the Condition Number

---

**Algorithm 2** AdaGrad (Duchi et al., 2010a; McMahan & Streeter, 2010) *(All operations on vectors are element-wise.)*

---

**Input**: #Iterations $T$, a set $\mathcal{K}$, $\boldsymbol{x}_1 \in \mathcal{K}$, stepsize $\eta$

**for** $t = 1 \ldots T$ **do**

  Receive: $\nabla f(\boldsymbol{x}_t)$

  Set: $\boldsymbol{\eta}_t = \dfrac{\eta}{\sqrt{\sum_{i=1}^t (\nabla f(\boldsymbol{x}_i))^2}}$

  Update: $\boldsymbol{x}_{t+1} = \Pi_{\mathcal{K}} (\boldsymbol{x}_t - \boldsymbol{\eta}_t \nabla f(\boldsymbol{x}_t))$ where $\Pi_{\mathcal{K}}$ is the projection onto $\mathcal{K}$.

**end for**

Output: $\bar{\boldsymbol{x}} = \frac{1}{T} \sum_{t=1}^T \boldsymbol{x}_t$.

---

**Algorithm 3** AdaGrad with Restart

---

**Input**: #Rounds $N$, $\boldsymbol{x}_0 \in \mathbb{R}^d$, upper bound on $\|\boldsymbol{x}_0 - \boldsymbol{x}^*\|_\infty$ as $D_\infty$, strong convexity $\mu$, smoothness $M$

Set: $\bar{\boldsymbol{x}}_0 = \boldsymbol{x}_0$

**for** $i = 1 \ldots N$ **do**

  Run Algorithm 2 to get $\bar{\boldsymbol{x}}_i$ with $T = 32d\frac{M}{\mu}$, $\boldsymbol{x}_1 = \bar{\boldsymbol{x}}_{i-1}$, $\mathcal{K} = \{\boldsymbol{x} : \|\boldsymbol{x} - \bar{\boldsymbol{x}}_{i-1}\|_\infty^2 \leq \frac{D_\infty^2}{4^{i-1}}\}$, $\eta = \frac{D_\infty/\sqrt{2}}{2^{i-1}}$

**end for**

Output: $\bar{\boldsymbol{x}}_N$.

---

**Theorem B.1.** *Let $\mathcal{K}$ be a hypercube with $\|\boldsymbol{x} - \boldsymbol{y}\|_\infty \leq D_\infty$ for any $\boldsymbol{x}, \boldsymbol{y} \in \mathcal{K}$. For a convex function $f$, set $\eta = \frac{D_\infty}{\sqrt{2}}$, then Algorithm 2 guarantees for any $\boldsymbol{x} \in \mathcal{K}$:*

$$\sum_{t=1}^T f(\boldsymbol{x}_t) - f(\boldsymbol{x}) \leq \sqrt{2dD_\infty^2 \sum_{t=1}^T \|\nabla f(\boldsymbol{x}_t)\|^2} \ . \tag{8}$$

**Theorem B.2.** *For a $\mu$ strongly convex and $M$ smooth function $f$, denote its unique minimizer as $\boldsymbol{x}^* \in \mathbb{R}^d$. Given $\boldsymbol{x}_0 \in \mathbb{R}^d$, assume that $\|\boldsymbol{x}_0 - \boldsymbol{x}^*\|_\infty \leq D_\infty$, then Algorithm 3 guarantees:*

$$\|\bar{\boldsymbol{x}}_N - \boldsymbol{x}^*\|_\infty^2 \leq \frac{D_\infty^2}{4^N} \ .$$

*Thus, to get a $\boldsymbol{x}$ such that $\|\boldsymbol{x} - \boldsymbol{x}^*\|_\infty^2 \leq \epsilon$, we need at most $32d\frac{M}{\mu} \log_4 \left( D_\infty^2/\epsilon \right)$ gradient calls.*

*Proof of Theorem B.2.* Consider round $i$ and assume $\mathcal{K}$ passed to Algorithm 2 is bounded w.r.t. $\ell_\infty$ norm by $D_{\infty_i}$. When $f$ is $\mu$-strongly convex and $M$ smooth, let $\boldsymbol{x} = \boldsymbol{x}^*$, Equation (8) becomes:

$$\sum_{t=1}^T f(\boldsymbol{x}_t) - f(\boldsymbol{x}^*) \leq \sqrt{2dD_{\infty_i}^2 \sum_{t=1}^T \|\nabla f(\boldsymbol{x}_t)\|^2} \leq \sqrt{4MdD_{\infty_i}^2 \sum_{t=1}^T (f(\boldsymbol{x}_t) - f(\boldsymbol{x}^*))} \ ,$$

where the second inequality is by the $M$ smoothness of $f$. This gives:

$$\sum_{t=1}^{T} f(\boldsymbol{x}_t) - f(\boldsymbol{x}^*) \leq 4MdD_{\infty_i}^2 \;.$$

Let $\bar{\boldsymbol{x}}_i = \frac{1}{T} \sum_{t=1}^{T} \boldsymbol{x}_t$ we have by the $\mu$-strong-convexity that:

$$\|\bar{\boldsymbol{x}}_i - \boldsymbol{x}^*\|_{\infty}^2 \leq \|\bar{\boldsymbol{x}}_i - \boldsymbol{x}^*\|^2 \leq \frac{2}{\mu}(f(\bar{\boldsymbol{x}}) - f(\boldsymbol{x}^*)) \leq \frac{2}{\mu}\frac{1}{T}\sum_{t=1}^{T}(f(\boldsymbol{x}_t) - f(\boldsymbol{x}^*)) \leq \frac{8MdD_{\infty_i}^2}{\mu T} \;. \tag{9}$$

Put $T = 32d\frac{M}{\mu}$ in Equation (9) we have that $\|\bar{\boldsymbol{x}}_i - \boldsymbol{x}^*\|_{\infty}^2 \leq \frac{D_{\infty_i}^2}{4}$. Thus, after each round, the $\ell_\infty$ distance between the update $\bar{\boldsymbol{x}}_i$ and $\boldsymbol{x}^*$ is shrinked by half, which in turn ensures that $\boldsymbol{x}^*$ is still inside the $\mathcal{K}$ passed to Algorithm 2 in the next round with $D_{\infty_{i+1}} = \frac{D_{\infty_i}}{2}$. This concludes the proof. $\qquad\square$

*Proof of Theorem B.1.*

$$\sum_{t=1}^{T} f(\boldsymbol{x}_t) - f(\boldsymbol{x})$$

$$\leq \sum_{t=1}^{T} \langle \nabla f(\boldsymbol{x}_t), \boldsymbol{x}_t - \boldsymbol{x} \rangle$$

$$= \sum_{t=1}^{T} \sum_{j=1}^{d} \frac{\partial f}{\partial x_{t,j}}(\boldsymbol{x}_t) * (x_{t,j} - x_j)$$

$$= \sum_{t=1}^{T} \sum_{j=1}^{d} \frac{(x_{t,j} - x_j)^2 - \left(x_{t,j} - \eta_{t,j}\frac{\partial f}{\partial x_{t,j}}(\boldsymbol{x}_t) - x_j\right)^2}{2\eta_{t,j}} + \sum_{t=1}^{T} \sum_{j=1}^{d} \frac{\eta_{t,j}}{2}\left(\frac{\partial f}{\partial x_{t,j}}(\boldsymbol{x}_t)\right)^2$$

$$\leq \sum_{t=1}^{T} \sum_{j=1}^{d} \frac{(x_{t,j} - x_j)^2 - (x_{t+1,j} - x_j)^2}{2\eta_{t,j}} + \sum_{t=1}^{T} \sum_{j=1}^{d} \frac{\eta_{t,j}}{2}\left(\frac{\partial f}{\partial x_{t,j}}(\boldsymbol{x}_t)\right)^2$$

$$\leq \sum_{j=1}^{d} \sum_{t=1}^{T} \frac{(x_{t,j} - x_j)^2}{2}\left(\frac{1}{\eta_{t,j}} - \frac{1}{\eta_{t-1,j}}\right) + \sum_{j=1}^{d} \sum_{t=1}^{T} \frac{\eta_{t,j}}{2}\left(\frac{\partial f}{\partial x_{t,j}}(\boldsymbol{x}_t)\right)^2$$

$$\leq \frac{D_\infty^2}{2\eta} \sum_{j=1}^{d} \sum_{t=1}^{T} \left(\sqrt{\sum_{i=1}^{t}\left(\frac{\partial f}{\partial x_{i,j}}(\boldsymbol{x}_i)\right)^2} - \sqrt{\sum_{i=1}^{t-1}\left(\frac{\partial f}{\partial x_{i,j}}(\boldsymbol{x}_i)\right)^2}\right) + \sum_{j=1}^{d} \sum_{t=1}^{T} \frac{\eta}{2\sqrt{\sum_{i=1}^{t}\left(\frac{\partial f}{\partial x_{i,j}}(\boldsymbol{x}_i)\right)^2}}\left(\frac{\partial f}{\partial x_{t,j}}(\boldsymbol{x}_t)\right)^2$$

$$\leq \sum_{j=1}^{d} \left(\frac{D_\infty^2}{2\eta}\sqrt{\sum_{t=1}^{T}\left(\frac{\partial f}{\partial x_{t,j}}(\boldsymbol{x}_t)\right)^2} + \eta\sqrt{\sum_{t=1}^{T}\left(\frac{\partial f}{\partial x_{t,j}}(\boldsymbol{x}_t)\right)^2}\right)$$

$$= \sum_{j=1}^{d} \sqrt{2D_\infty^2 \sum_{t=1}^{T}\left(\frac{\partial f}{\partial x_{t,j}}(\boldsymbol{x}_t)\right)^2}$$

$$\leq \sqrt{2dD_\infty^2 \sum_{t=1}^{T}\sum_{j=1}^{d}\left(\frac{\partial f}{\partial x_{t,j}}(\boldsymbol{x}_t)\right)^2}$$

$$= \sqrt{2dD_\infty^2 \sum_{t=1}^{T}\|\nabla f(\boldsymbol{x}_t))\|^2} \;.$$

where the first inequality is by convexity, the second one by the projection lemma as the projection onto a hypercube equals performing the projection independently for each coordinate, the fifth one by Lemma 5 in (McMahan & Streeter, 2010), and the last one by the concavity of $\sqrt{\cdot}$. $\qquad\square$

## C  The Histograms of the Magnitude of each Update Coordinate during the Entire Training Phase

In this section, we report the histograms of the absolute value of updates of Adam-$\ell_2$ vs. AdamW of all coordinates divided by $\alpha$ during the whole training process. From the figures shown below, we can clearly see that AdamW's updates remain in a much more concentrated scale range than Adam-$\ell_2$ during the entire training. Moreover, as the depth of the network grows, Adam-$\ell_2$'s updates become more and more dispersed, while AdamW's updates are still concentrated. *(Note that the leftmost bin contains all values equal to or less than $2^{-27} \approx 10^{-8.1}$ and the rightmost bin contains all values equal to or larger than $1$.)*

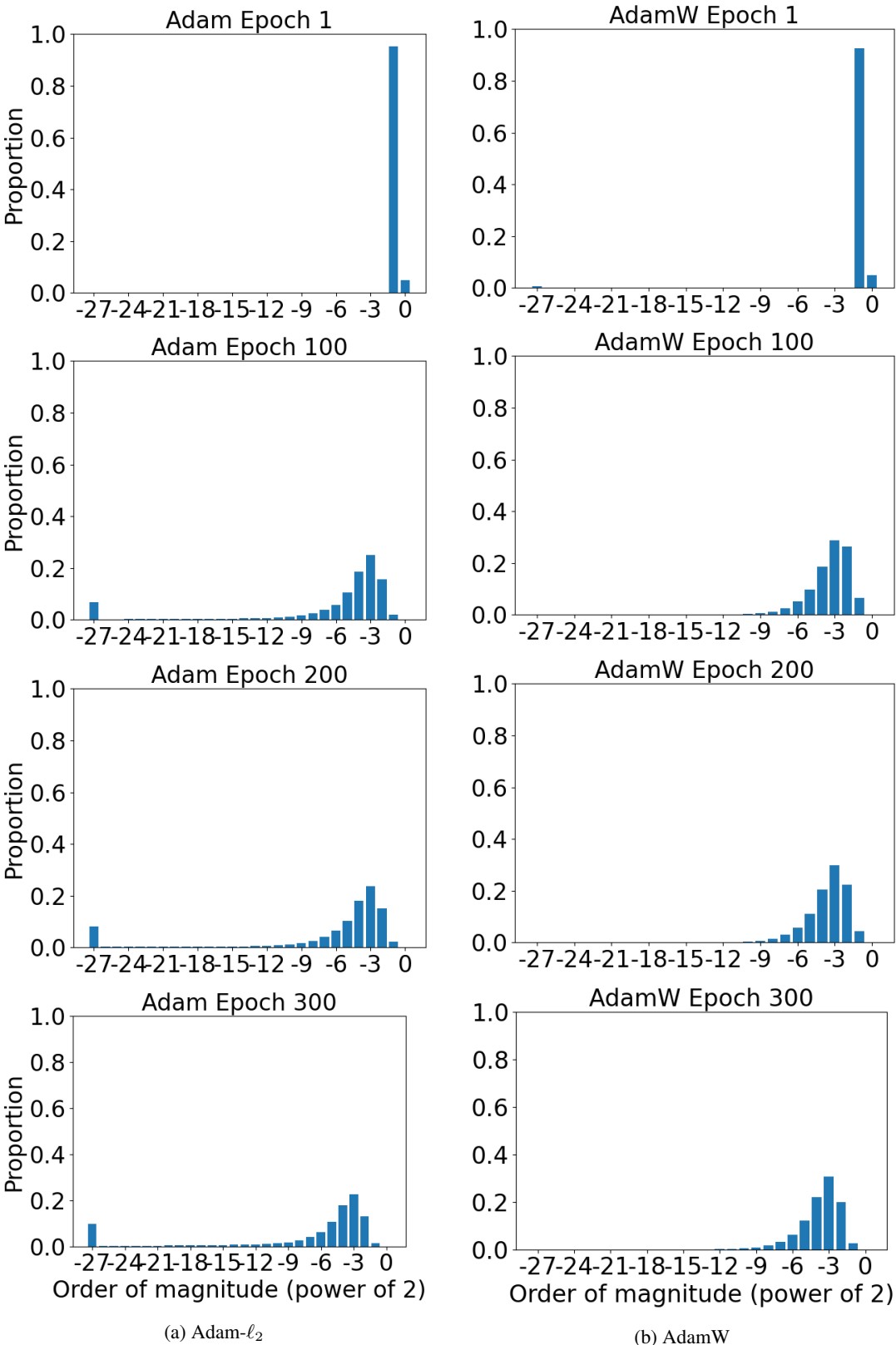

(a) Adam-$\ell_2$

(b) AdamW

Figure 8: The histograms of the magnitudes of all updates (without $\alpha$) of a 20-layer Resnet with BN removed trained by AdamW or Adam-$\ell_2$ on CIFAR10.

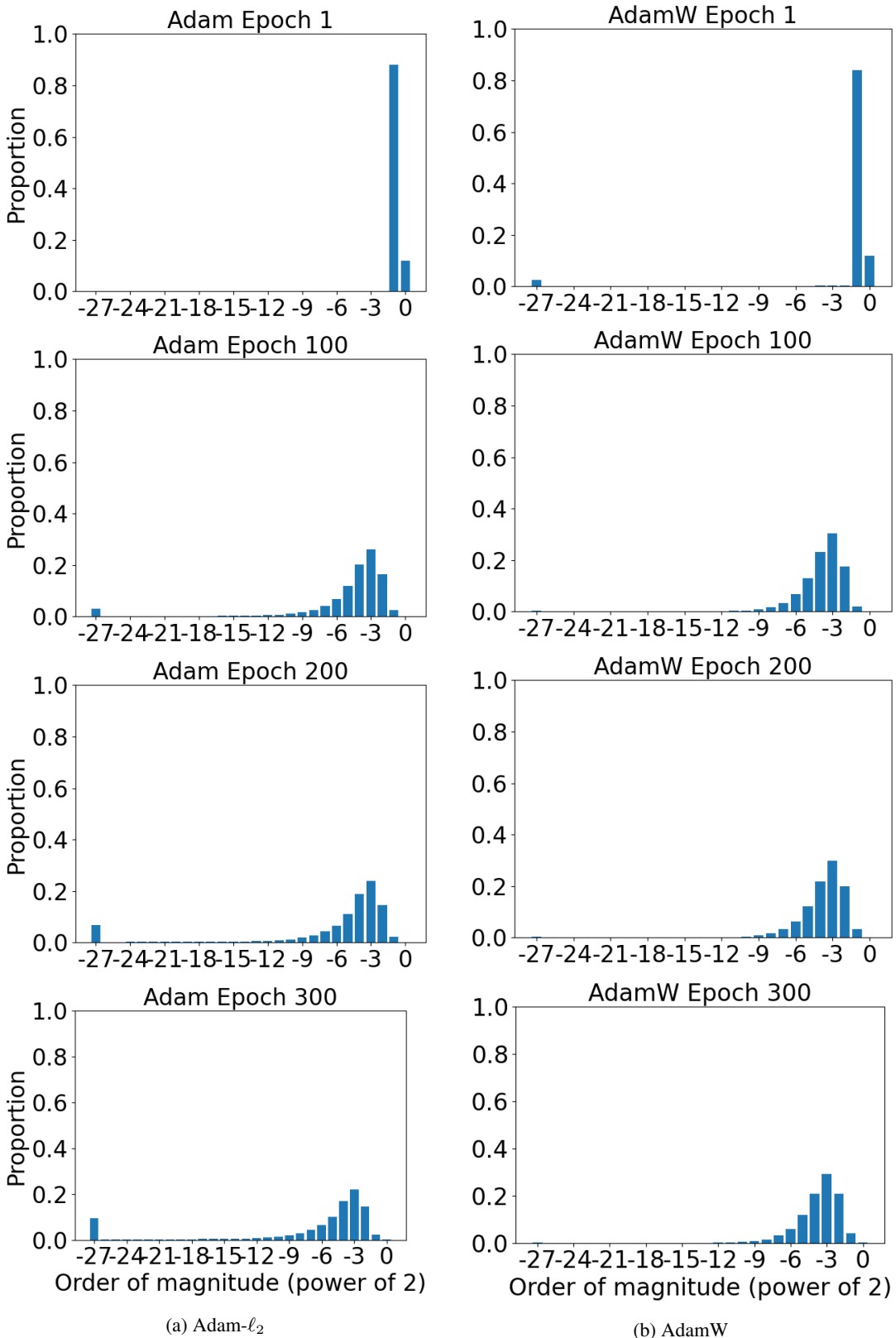

(a) Adam-$\ell_2$

(b) AdamW

Figure 9: The histograms of the magnitudes of all updates (without $\alpha$) of a 44-layer Resnet with BN removed trained by AdamW or Adam-$\ell_2$ on CIFAR10.

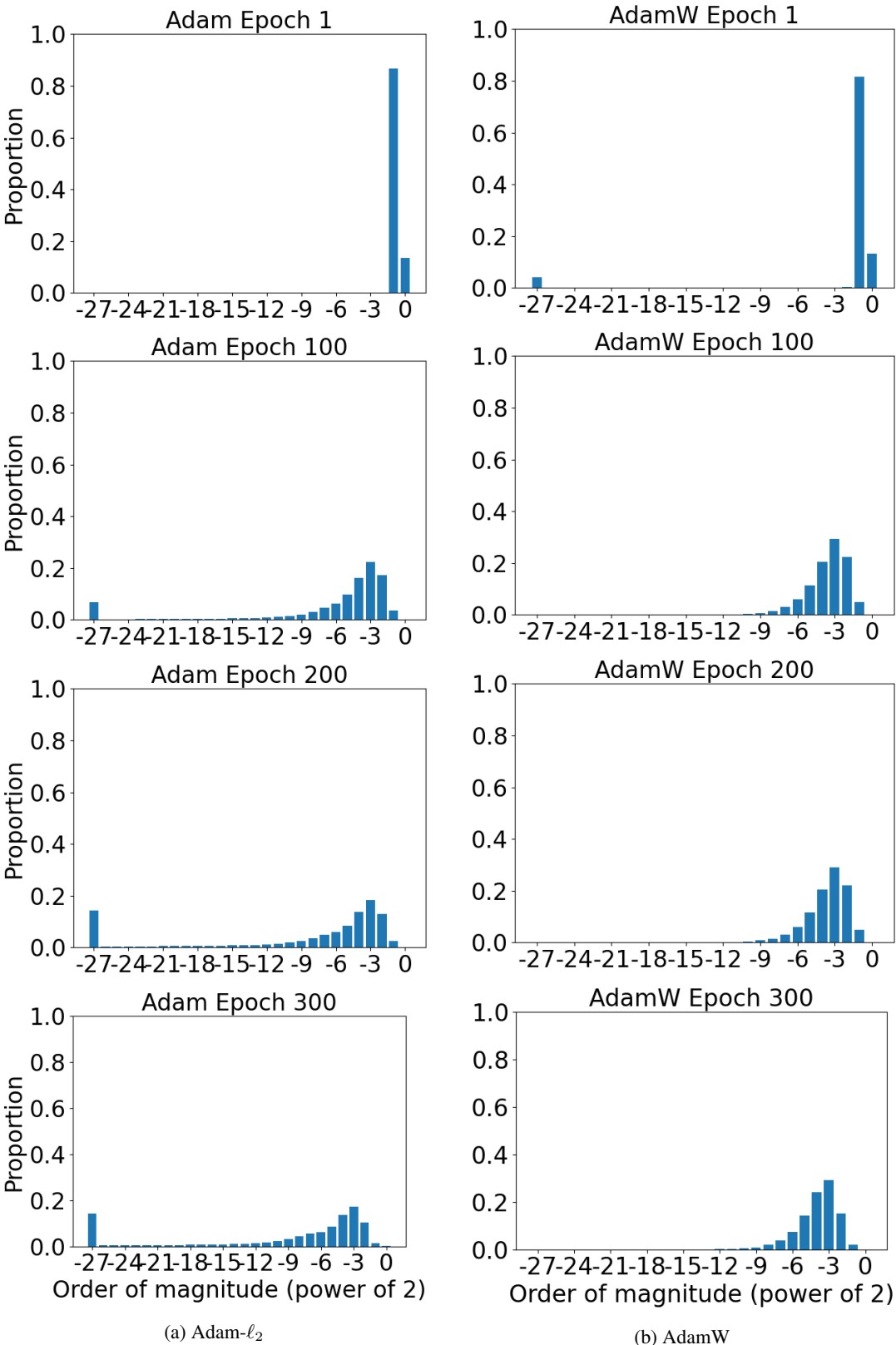

(a) Adam-$\ell_2$

(b) AdamW

Figure 10: The histograms of the magnitudes of all updates (without $\alpha$) of a 56-layer Resnet with BN removed trained by AdamW or Adam-$\ell_2$ on CIFAR10.

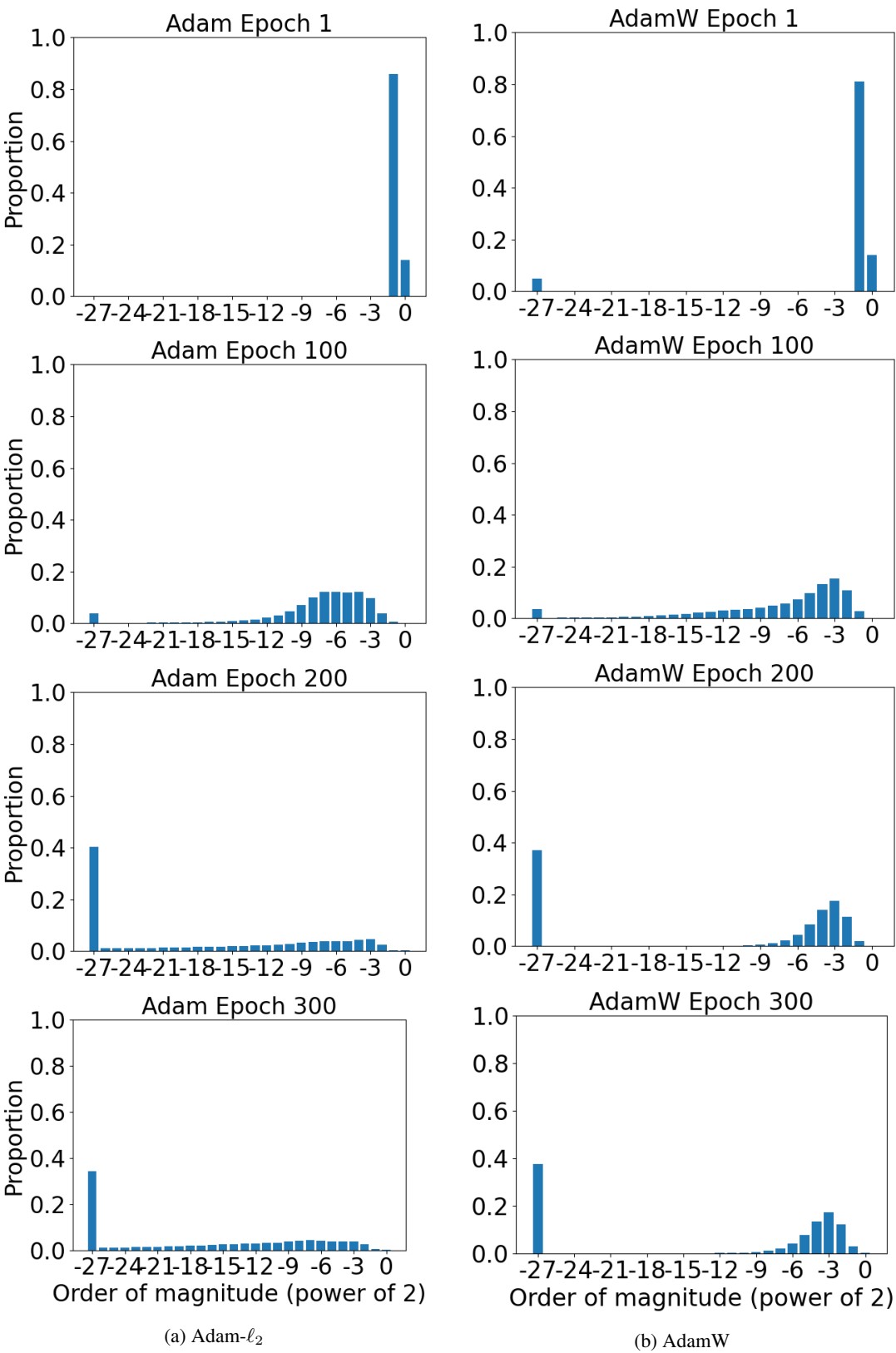

(a) Adam-$\ell_2$

(b) AdamW

Figure 11: The histograms of the magnitudes of all updates (without $\alpha$) of a 110-layer Resnet with BN removed trained by AdamW or Adam-$\ell_2$ on CIFAR10.

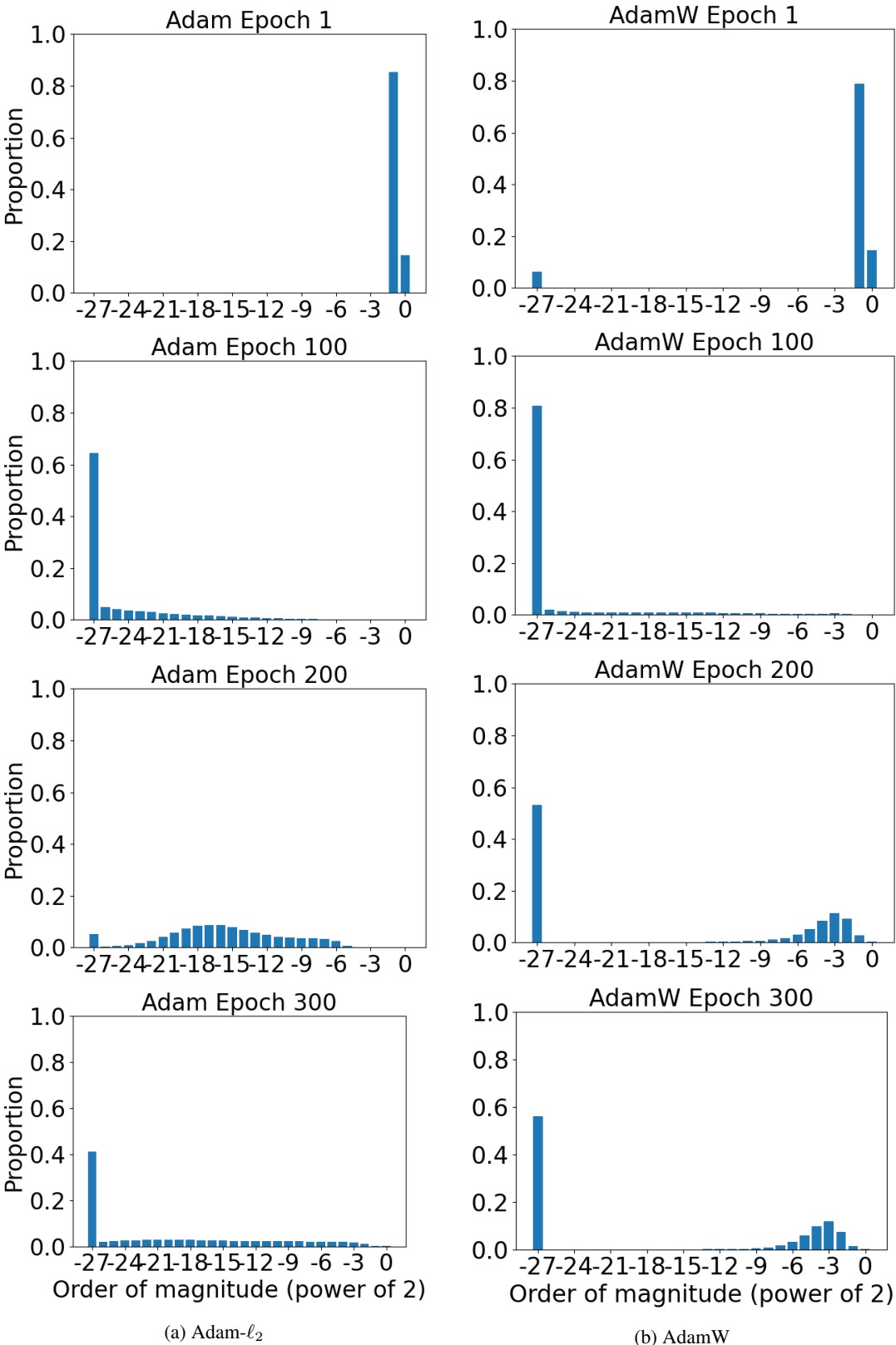

(a) Adam-$\ell_2$

(b) AdamW

Figure 12: The histograms of the magnitudes of all updates (without $\alpha$) of a 218-layer Resnet with BN removed trained by AdamW or Adam-$\ell_2$ on CIFAR10.

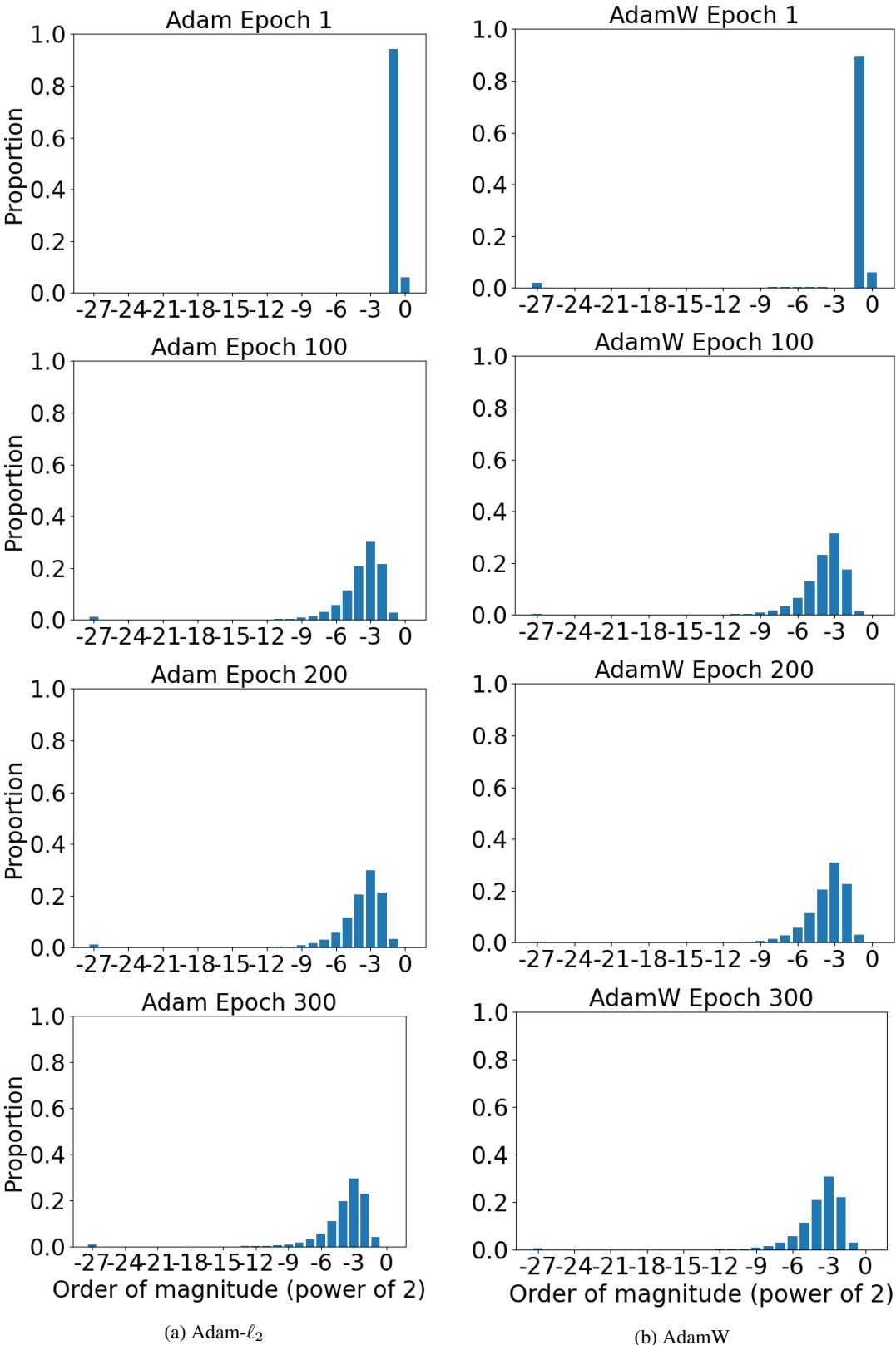

(a) Adam-$\ell_2$

(b) AdamW

Figure 13: The histograms of the magnitudes of all updates (without $\alpha$) of a 20-layer Resnet with BN removed trained by AdamW or Adam-$\ell_2$ on CIFAR100.

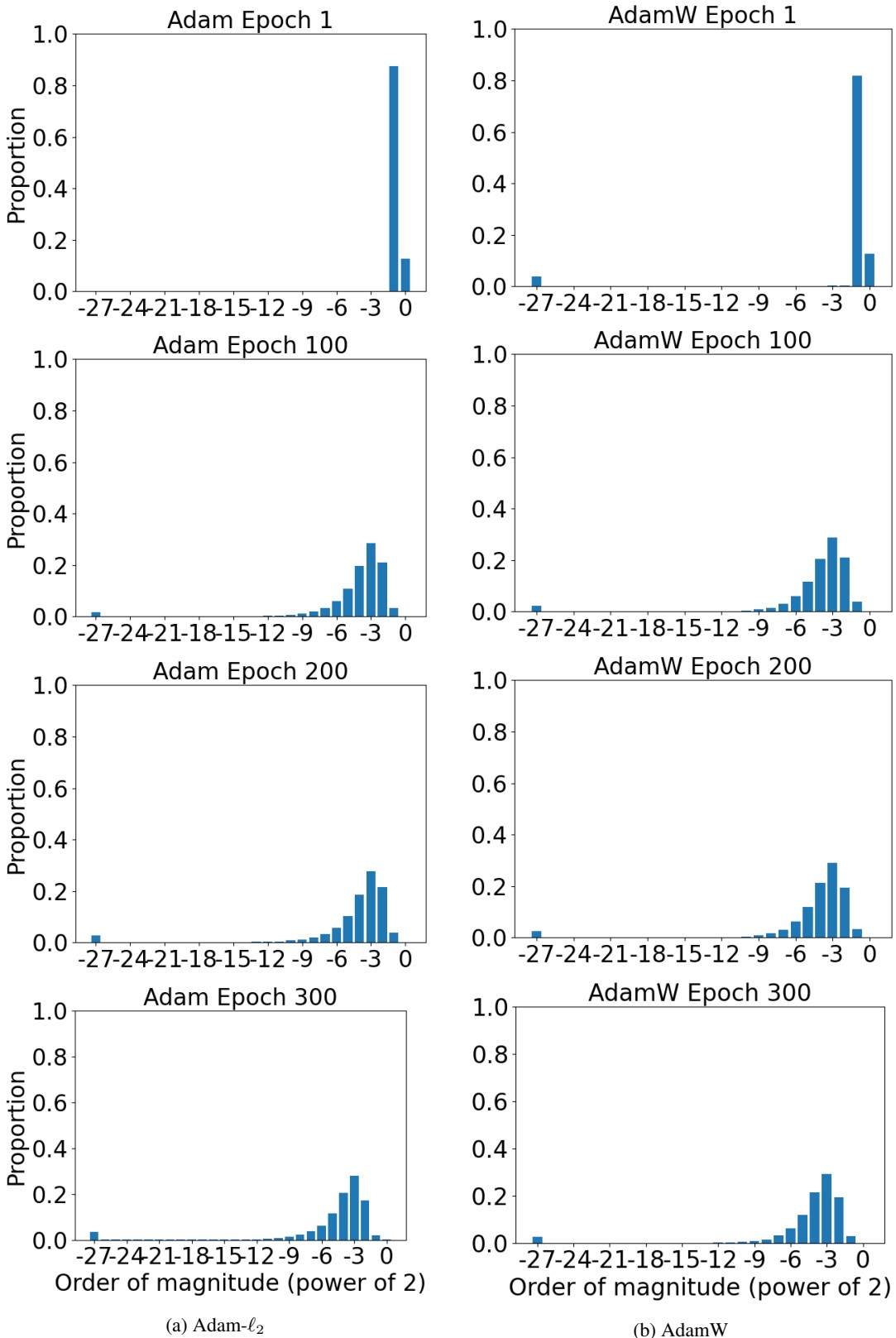

(a) Adam-$\ell_2$

(b) AdamW

Figure 14: The histograms of the magnitudes of all updates (without $\alpha$) of a 44-layer Resnet with BN removed trained by AdamW or Adam-$\ell_2$ on CIFAR100.

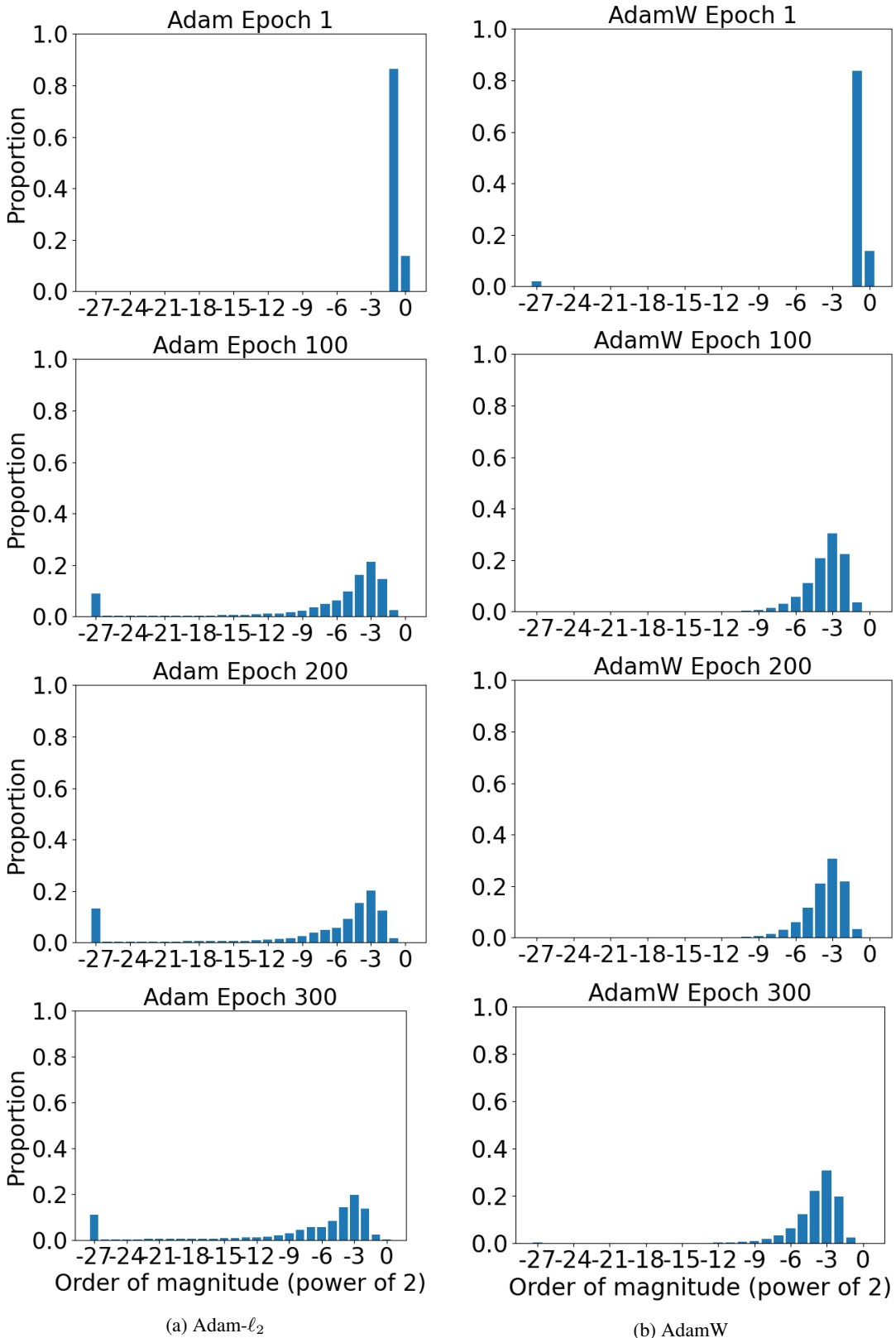

(a) Adam-$\ell_2$

(b) AdamW

Figure 15: The histograms of the magnitudes of all updates (without $\alpha$) of a 56-layer Resnet with BN removed trained by AdamW or Adam-$\ell_2$ on CIFAR100.

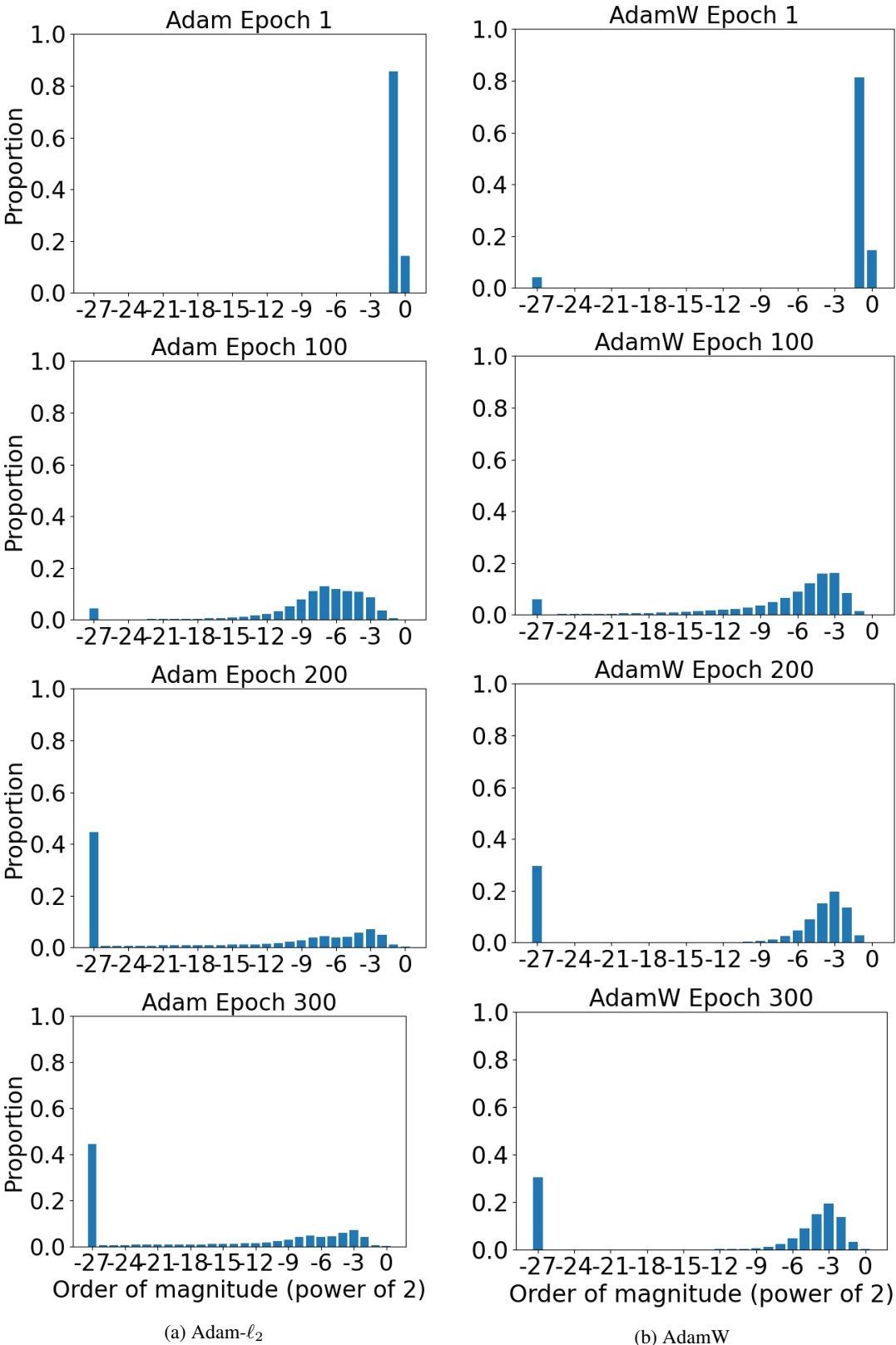

(a) Adam-$\ell_2$

(b) AdamW

Figure 16: The histograms of the magnitudes of all updates (without $\alpha$) of a 110-layer Resnet with BN removed trained by AdamW or Adam-$\ell_2$ on CIFAR100.

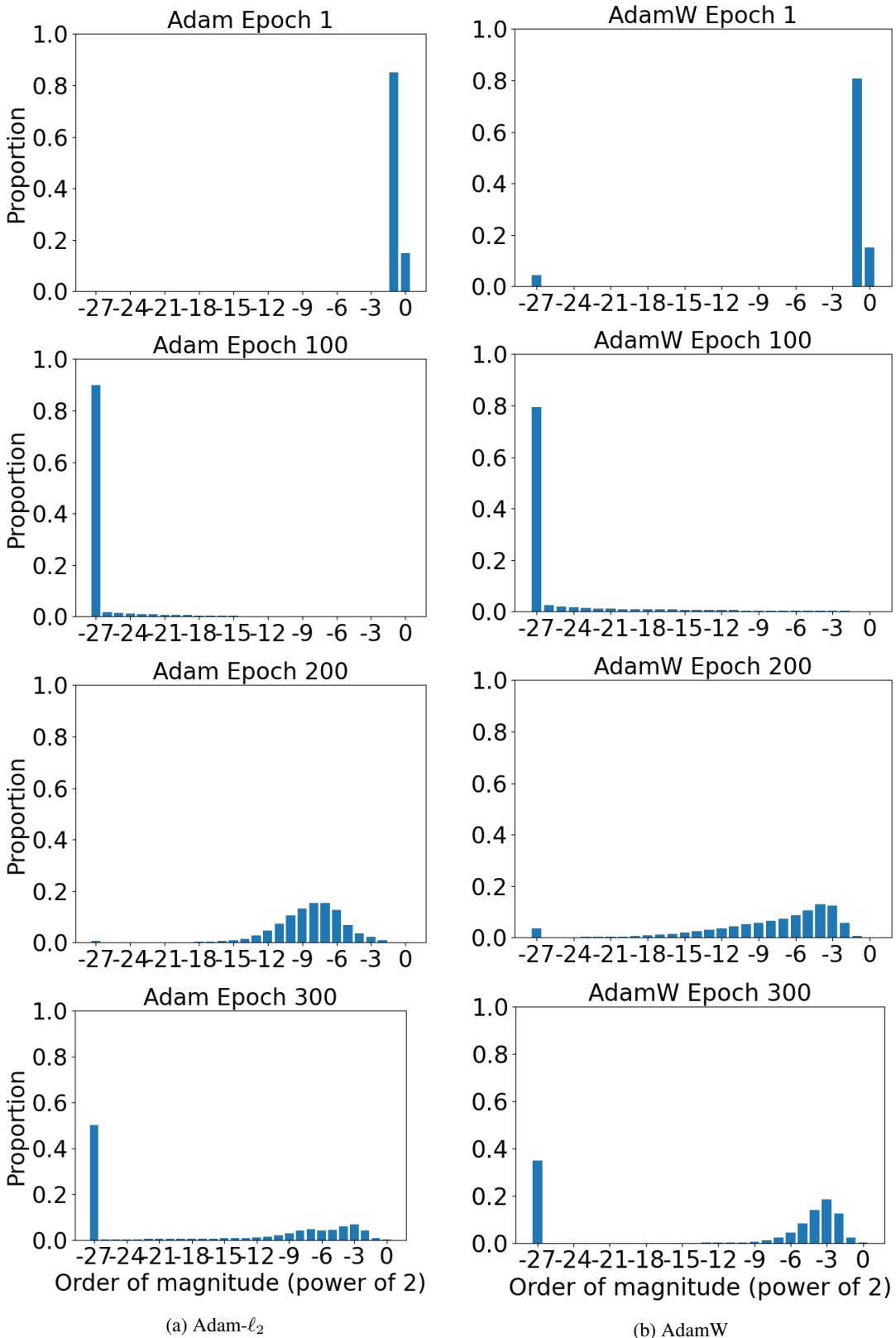

Figure 17: The histograms of the magnitudes of all updates (without $\alpha$) of a 218-layer Resnet with BN removed trained by AdamW or Adam-$\ell_2$ on CIFAR100.

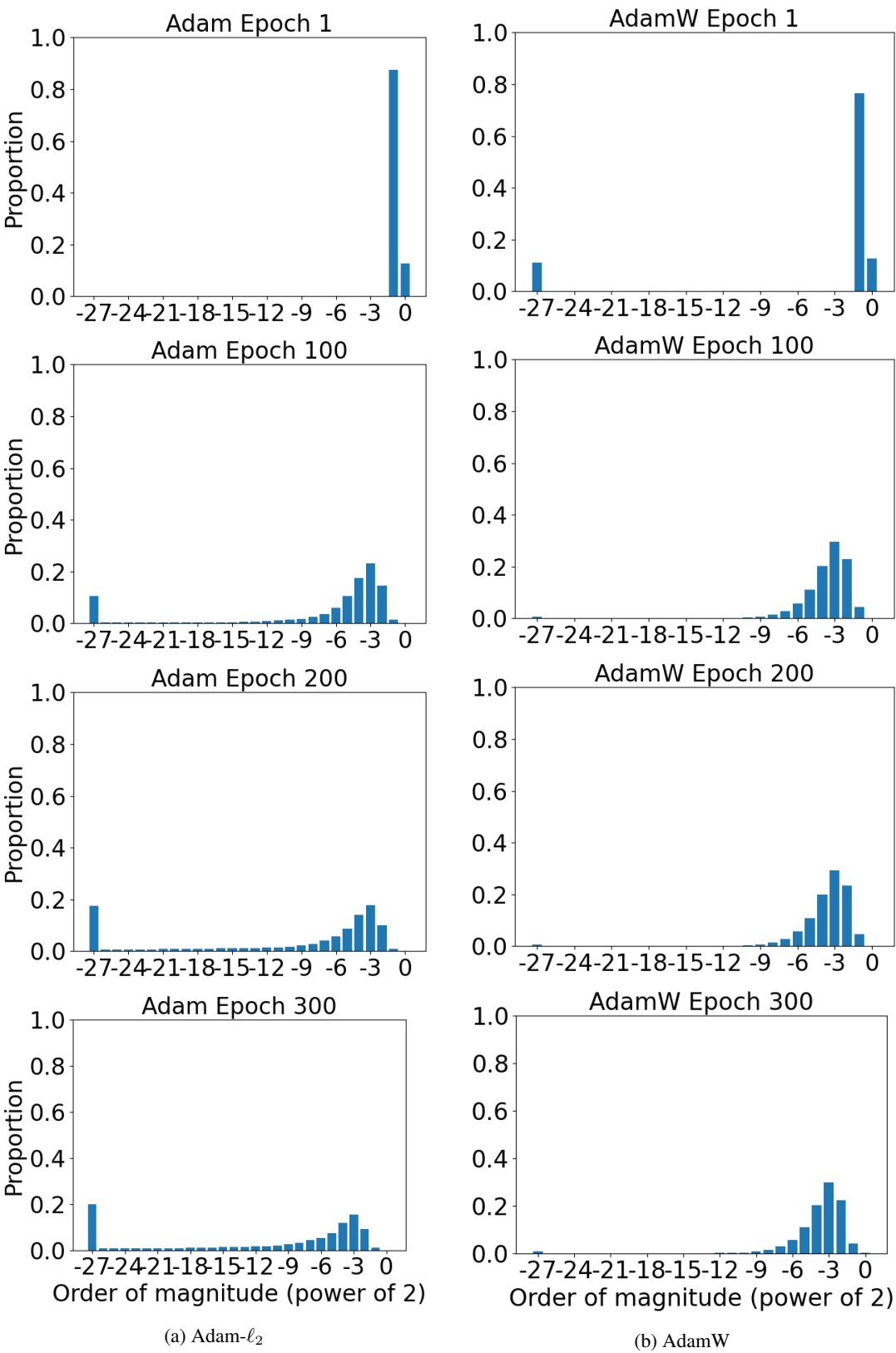

(a) Adam-$\ell_2$

(b) AdamW

Figure 18: The histograms of the magnitudes of all updates (without $\alpha$) of a 100-layer DenseNet-BC with BN removed trained by AdamW or Adam-$\ell_2$ on CIFAR100.

