# OpenReview forum: "Understanding AdamW through Proximal Methods and Scale-Freeness"
_TMLR — Accepted by TMLR_

### Review · Reviewer_i3NP · 2022-06-17

**Summary Of Contributions:**

The manuscript investigates the benefits of AdamW by deriving connections to proximal methods and showing it exhibits a scale-freeness property. The paper also provides empirical validation of some approximations made in the derivation and demonstrates that the effects hypothesised by the theoretical analysis are observed in practice.

**Broader Impact Concerns:**

No concerns.

**Requested Changes:**

Some of the figures in Section 4 have plots that have inconsistent sizes. It would be nice (but not required for acceptance) if these could be fixed, and perhaps slightly more guidance could be given to the reader about what features of the plots can be used to draw the conclusions the authors have drawn.

The formalisation at the beginning of Section 3 should be improved before acceptance. A touch more scene-setting would be useful; e.g., introduce the definition of a proximal operation, and use it to set up some notation that allows for compact and easily understandable derivations. Doing this may also alleviate what I thought was the main weakness of the paper, by making the relationship to existing proximal methods more clear.

**Strengths And Weaknesses:**

For the most part, I found the paper quite easy to follow and enjoyable to read. The theoretical contributions are quite interesting, and have the potential (as evidenced by the planned future work) to lead to new training algorithms with similarly beneficial properties. The experimental work did a good job of demonstrating the practical implications of scale-freeness in the deep learning setting.

The main weakness is that the term "proximal update" is not always used in a precise manner, so it is at times unclear which of the existing proximal methods the manuscript is referring to. For example, the manuscript states that Eq 4 "uses the proximal operator", but it seems to me that there are additional terms compared to a conventional proximal operation definition, and a different Bregman divergence is used. Making this clear is important, as the manuscript does make some claims about how AdamW inherits properties that pertain to specific proximal methods. It is not clear to me how the momentum terms from Adam(W) interact with this proximal update interpretation. E.g., the convergence guarantees of Duchi et al. (2010b) are invoked, but it is not obvious to me that they apply in the case where momentum is present.

---

### Review · Reviewer_ZnRb · 2022-06-17

**Summary Of Contributions:**

The paper provides some theoretical and empirical insights on AdamW, especially on why it can outperform Adam l2. The key messages can be summarized in two points. 1) AdamW can be viewed as a first-order approximation of a special proximal method. 2) The advantage of AdamW over Adam l2 might be brought by its scale-freeness property.

Overall I enjoyed reading the paper and the idea is interesting. But I do have a few questions for the authors.

1. What is the main further insight brought by interpreting AdamW as an approximation of proximal methods?

2. Another intuitive understanding of the advantages of AdamW over Adam l2 is treating AdamW as assigning different l2 regularization parameters to different coordinates, according to the gradient magnitudes of different coordinates. The advantage then would be AdamW is regularizing the l2 norm of different weights according to their importance in changing the objective function. I believe this is a more common and easily understood perspective than scale-freeness. I wonder how does the authors view this perspective and are there connections between scale-freeness and this perspective?

**Broader Impact Concerns:**

I do not have concerns.

**Requested Changes:**

It seems the paper is mainly on scale-freeness and the connection to proximal methods does not provide much insight. Also, it feels to me that scale-freeness and the connection to proximal methods are separated results without much connection to each other.  Maybe it is better to restructure the paper to highlight scale-freeness. Or add some transition between the connection to proximal methods and the scale-freeness part.

**Strengths And Weaknesses:**

Strength is the perspective is new to the best of my knowledge.
Weakness is the limited theoretical depth.

---

### Review · Reviewer_enES · 2022-06-18

**Summary Of Contributions:**

This paper studies the advantage of AdamW over Adam, which is a well-known phenomenon especially in computer vision tasks, where people typically argue that AdamW is better because its implementation of weight decay is correct while Adam's is not. This paper provides a different perspective of viewing the two methods in terms of whether scale-freeness is satisfied, and show both theoretically and empirically that this property helps AdamW to outperform Adam. Extensive experimental results are provided by the authors. The proposed ideas are, as far as I know, novel and interesting. However, I kind of have the feeling that scale-freeness is another way of saying "weight decay" is better than "l_2 regularization" in adaptive methods.

**Broader Impact Concerns:**

This paper studies optimization algorithms and I don't see any concerns with the ethical implications.

**Requested Changes:**

Overall, this paper is a sound empirical work and I would recommend for publication if the following problems are fixed or addressed properly.

Page 4, the paragraph before "Adam is Scale Free" is very unclear to me. The authors refer to different cases of $x_t$, different cases of $g_t$ and many update rules including (2) and (5). Moreover the term $p_t$, as in the paper, can have different forms. Therefore, I am really confused when the author says "when $g_t$ is zero". Even if $g_t$ is zero, what is $p_t$ in these cases? I would strongly suggest that the author make a table for the two cases they mention, and what the consequences will be in different cases. With Eqn. (2) and (5) in different pages, I have to go back and forth when reading this paragraph and I am still confused by what the authors really mean.

Some notations are never defined. For example, what is $m_t$ and what is $f_t$ in page 3/4? As someone who studies optimization, I guess the author means the "first order momentum" and the "stochastic approximation" to the loss function $f$. However, please define them clearly if theoretical analysis is provided. " with $m_t$ and $v_t$ both updated using $g_t$" is very vague. Please write out the update rules of Adam directly.

For the experiments, I am surprised that when BN is turned on, AdamW and Adam behaves the best when the regularization is zero since it will lead to overfitting when I run them. However, I am even more surprised by Figure 2(b). When both Adam and AdamW have zero weight decay, shouldn't them be the same? Why is the best learning rate 1 for Adam and 5 for AdamW? Could the authors explain it to me?

As I mentioned above, it is better if scale-free SGD is added to the comparison, but this is less important.

Minor errors: some citations are used incorrectly, e.g., Duchi et al. 2010b in page 4, Nesterov 2004 in page 5. I believe the authors want to use \citep instead of \citet there.





**Strengths And Weaknesses:**

Strengths:
1. The proposed idea is novel and interesting.
2. The authors have conducted numerous experiments to verify their claims.
3. The paper has nice comparisons between Adam, AdamW, and AdamProx(as proposed in the paper)

Weaknesses:
1. One part of the paper is very unclear (see below)
2. The "theory" part of this paper is rather intuitive than explanatory, only the strongly convex setting is considered
3. I do not see the impact of this work, or how it can help us design the optimizers. Since scale-freeness is a desired property, how about we design a scale-free version of SGD, e.g., scale the gradients by their norms. How is that algorithm going to perform on neural networks without BN? Since the authors only compare Adam with AdamW, and I guess SGD will likely fail in those networks, I wonder how the scaled-free version of SGD is going to perform.

---

### Decision · Action_Editors · 2022-07-26

**Recommendation:** Accept as is

**Comment:**

In this paper, the authors provide new theoretical understandings of AdamW from the optimization viewpoint. First, they demonstrate that AdamW can be seen as an approximation of a proximal update. Second, they prove that AdamW enjoys the scale-freeness property, which leads to an automatic acceleration on certain cases.

This paper made significant progress towards understanding the advantage of AdamW. It is well-written, and the theoretical findings are supported by empirical studies.

---

> ### Author Response · Authors · 2022-08-12
> **Thank you!**
>
> We would like to express our most sincere gratitude to the action editor and all the reviewers for your time and efforts!